# On the Relation between Rectified Flows and Optimal Transport

**Johannes Hertrich**
Université Paris Dauphine - PSL
& Inria Mokaplan, Paris
`johannes.hertrich@dauphine.psl.eu`

**Antonin Chambolle**
Université Paris Dauphine - PSL
& Inria Mokaplan, Paris
`chambolle@ceremade.dauphine.fr`

**Julie Delon**
ENS Paris
`julie.delon@ens.fr`

## Abstract

This paper investigates the connections between rectified flows, flow matching, and optimal transport. Flow matching is a recent approach to learning generative models by estimating velocity fields that guide transformations from a source to a target distribution. Rectified flow matching aims to straighten the learned transport paths, yielding more direct flows between distributions. Our first contribution is a set of invariance properties of rectified flows and explicit velocity fields. In addition, we also provide explicit constructions and analysis in the Gaussian (not necessarily independent) and Gaussian mixture settings and study the relation to optimal transport. Our second contribution addresses recent claims suggesting that rectified flows, when constrained such that the learned velocity field is a gradient, can yield (asymptotically) solutions to optimal transport problems. We study the existence of solutions for this problem and demonstrate that they only relate to optimal transport under assumptions that are significantly stronger than those previously acknowledged. In particular, we present several counterexamples that invalidate earlier equivalence results in the literature, and we argue that enforcing a gradient constraint on rectified flows is, in general, not a reliable method for computing optimal transport maps.

## 1 Introduction

Optimal transport is the problem of transporting a probability distribution $\mu_0$ to another distribution $\mu_1$ such that a given cost function has minimal expected value. More precisely, we aim to find a coupling $(X_0, X_1)$ of random variables $X_0 \sim \mu_0$ and $X_1 \sim \mu_1$ such that $\mathbb{E}[c(X_0, X_1)]$ is minimal, where $c$ is a cost function, most commonly $c(x, y) = \|x - y\|$ (1-*Wasserstein or earth-mover distance*) or $c(x, y) = \|x - y\|^2$ (*squared* 2-*Wasserstein distance*). The problem was originally formulated by Monge [39] in terms of optimal transport maps and later generalized by Kantorovich [25] using more general couplings. We refer to [42, 45, 51] for an overview. Nowadays, it is heavily used in machine learning for clustering [23, 38], domain adaptation [12], generative modeling [5, 27] or model selection [6], to cite just a few references.

Computing solutions to the optimal transport problem can be very challenging in practice. In the discrete case, it amounts to solving a linear program and can be efficiently accelerated using entropic regularization and the Sinkhorn algorithm [13, 47]. A dynamic formulation of optimal transport, introduced by Benamou and Brenier [7], characterizes optimal solutions as those induced by a velocity field $v_t$ of minimal $L^2$-norm (for the 2-Wasserstein distance) among fields transporting $\mu_0$ to $\mu_1$ via

39th Conference on Neural Information Processing Systems (NeurIPS 2025).

the ordinary differential equation $\dot{z}_t(x) = v_t(z_t(x))$. From an analytic viewpoint this corresponds to finding a curve $t \mapsto \mu_t$ interpolating between $\mu_0$ to $\mu_1$ in the space of probability measures, such that the continuity equation $\partial_t \mu_t + \mathrm{div}(v_t \mu_t) = 0$ holds and $\int_t \int \|v_t\|^2 d\mu_t dt$ is minimal.

This dynamic perspective on transporting probability measures is also used in flow-based generative models, such as continuous normalizing flows [11, 19] or denoising diffusion models [22, 48]. However, the transport maps learned by these models are in general not optimal.

Recently, Liu et al. [35] proposed to learn such a velocity field by starting from an arbitrary coupling and averaging up all linear interpolations between $X_0$ and $X_1$. Similar ideas were introduced at the same time in [2, 32] under the name flow matching and stochastic interpolants. Building on this framework, rectified flow matching seeks to straighten the learned transport paths, leading to more direct flows between distributions.

**Outline and Contributions**   In this paper, we study the relation between rectified flows and optimal transport. In particular, we show that the iterative rectification proposed in [34] is not, in general, a suitable tool for computing optimal transport maps. In Section 2, we start with revisiting the backgrounds on rectified flows. Additionally, we show some invariance properties under affine transformations and derive the optimal velocity fields for the Gaussian case. Then, we show in Section 3 how these properties change if we constrain the velocity fields to be a gradient. In particular, we prove that a solution of the constrained problem always exists in a weak form. Afterwards, in Section 4, we construct the following two counterexamples, where the relation between optimal transport couplings and fixed points of the rectification procedure is false :

- First, in Section 4.1, we consider an example where the support of the interpolated distributions $\mu_t$ is disconnected. In this case, we can divide the space in several disconnected subdomains. Then, we can construct a transport which is optimal on each subdomain, but not globally optimal. This example shows in particular, that the claim from [34, Theorem 5.6] is not true without additionally assuming that $X_t$ has connected support. Since datasets in applications are often disconnected this massively restricts the applicability of rectified flows for computing the optimal transport.

- Second, we pay attention to so-called non-rectifiable couplings in Section 4.2. These are couplings such that the velocity field $v_t$ learned by the rectification does not lead to a unique solution of the ODE $\dot{Z}_t = v_t(Z_t)$. We show an example that such couplings exist and can have zero loss even though they are not optimal.

We prove a sufficient criterion for a coupling to be rectifiable based on the smoothness of the conditional probability of $X_0$ given $X_1$. This criterion is in particular fulfilled for the independent coupling with smooth initial distribution $\mu_0$ or when a small amount of noise is injected into $X_0$. In Section 5 we consider the special case where $\mu_0$ is a Gaussian distribution. Then, we modify the iterative rectification by injecting noise in each step to ensure that the iterates remain rectifiable. We show that the arising procedure is still marginal preserving and has the same decay of the loss function as without noise injection. We discuss the implications of our results and draw conclusions in Section 6.

**Related Work**   Rectified flows or flow matching was introduced in [2, 32, 35] and further investigated in [1], see [33, 52] for an overview. Error of the generated distribution by the training error in the velocity field were proven by [8, 44]. The basic idea of defining an interpolation path of latent and target measure was previously used in diffusion models and several other models [20, 53]. Applications and improvements of the training procedure of rectified flows were considered in [10, 29, 30, 36, 37].

In order to compute the optimal transport using a flow-based generative model, [21, 40, 54] include a regularization term of the velocity field into the loss function of a continuous normalizing flow [11, 19]. However, this approach alters the marginals of the solution. Other papers [24, 26] propose to compute optimal transport maps by using Brenier's theorem [9] and representing the convex potential of the transport map by an input convex neural network. The authors of [28] use the dual formulation of optimal transport and solve the arising saddle point problem with neural networks. The authors of [1, 2] relate flow matching with (entropic) optimal transport by considering non-linear interpolants between the marginals.

To initialize rectified flows close to the optimal coupling, [43, 49] propose minibatch OT. That is, they draw a minibatches from both marginals $\mu_0$ and $\mu_1$ and pair the data points within these batches by computing the discrete optimal transport between them. However, the coupling generated by minibatch OT is again not optimal in general.

In [1] the authors propose a noisy version of flow matching. Iterating this procedure is related to the Schrödinger bridge problem under the name Diffusion Schrödinger Bridge Matching (DSBM) [14, 46]. This corresponds to computing the entropically regularized optimal transport plan, see [31] for an overview. While convergence is guaranteed for DSBM under mild assumptions, our counterexamples suggest that the convergence rate becomes arbitrary slow when the entropic regularization parameter tends to zero. We include some numerical tests in this direction in Appendix F.

## 2 Rectified Flows

In this section, we provide an overview of the backgrounds of rectified flows. Afterwards, we derive invariance properties of the rectification procedure and study the case where all involved measures are (mixtures of) Gaussians. We will see that these invariance properties already have some similarities to optimal transport. Additionally, they will be needed in the proofs later in the paper.

### 2.1 Backgrounds

**Wasserstein Distance**   We define a coupling between two probability measures $\mu_0$ and $\mu_1$ on $\mathbb{R}^d$ as a pair $(X_0, X_1)$ of random variables with $X_0 \sim \mu_0$ and $X_1 \sim \mu_1$. The Wasserstein-2 distance is given by

$$W_2^2(\mu_0, \mu_1) = \inf_{X_0 \sim \mu_0, X_1 \sim \mu_1} \mathbb{E}[\|X_0 - X_1\|^2].$$

In addition we say that a coupling $(X_0, X_1)$ between $\mu_0$ and $\mu_1$ is optimal if it fulfills $W_2^2(\mu_0, \mu_1) = \mathbb{E}[\|X_0 - X_1\|^2]$. The Wasserstein distance $W_2$ defines a metric on the space of probability measures with finite second moment. Throughout the paper, we assume that all considered probability measures belong to this space. The distance $W_2$ belongs to the family of optimal transport discrepancies, which are defined in the same way by replacing the squared Euclidean norm by some more general cost function $c$.

**Rectified Flows**   In order to build a generative model for some target measure $\mu_1$ based on some latent measure $\mu_0$, the authors of [34, 35] propose rectified flows, which are also known by the name flow matching [32, 33] or stochastic interpolants [1, 2]. Given a coupling $(X_0, X_1)$ between $\mu_0$ and $\mu_1$, we consider the interpolations $X_t = (1-t)X_0 + tX_1$ and denote by $\mu_t$ the distribution of $X_t$. Then, we construct a velocity field $(v_t)_{t \in [0,1]}$ for $\mu_t$ by minimizing the loss function

$$v_t \in \operatorname*{arg\,min}_{w_t \in L^2(\mu_t)} \mathcal{L}(w_t | X_0, X_1), \quad \mathcal{L}(w_t | X_0, X_1) := \int_0^1 \mathbb{E}[\|w_t(X_t) - X_1 + X_0\|^2]dt. \quad (1)$$

The authors of [2, 32, 35] show that the minimizer of this problem exists and is unique. Using the optimal prediction property of conditional expectations, the solution $v_t$ of (1) can be formulated as the conditional expectation

$$v_t(x) = \mathbb{E}[X_1 - X_0 | X_t = x] = \frac{1}{1-t}\mathbb{E}[X_1 - X_t | X_t = x] = \frac{1}{1-t}\left(\mathbb{E}[X_1 | X_t = x] - x\right). \quad (2)$$

Additionally, they show that the solution $v_t$ fulfills the continuity equation with respect to $\mu_t$, i.e., that

$$\partial_t \mu_t + \operatorname{div}(v_t \mu_t) = 0 \quad (3)$$

in a distributional sense. In particular, under the assumption that $v_t$ is smooth enough, $v_t$ defines a transport from $\mu_0$ to $\mu_1$ in the sense that $\mu_1 = z_{1\#}\mu_0$, where $z_t(x)$ is the solution of the ODE

$$\dot{z}_t(x) = v_t(z_t(x)), \quad \text{with initial condition} \quad z_0(x) = x. \quad (4)$$

If the ODE (4) has a unique solution, we can sample from $\mu_1$ by sampling from $\mu_0$ and solving the ODE (4). In the literature, the arising generative model produces state-of-the-art results [2, 32, 35].

**Iterative Rectification**   In order to obtain simpler velocity fields, Liu et al. [35] propose to construct a new "rectified" coupling as follows.

**Definition 1.** Let $(X_0, X_1)$ be a coupling between $\mu_0$ and $\mu_1$ and denote by $v_t$ the minimizer of the loss function (1). Then, we call $(X_0, X_1)$ "rectifiable" if the ODE (4) has a unique solution and define $(Z_0, Z_1) = \mathcal{R}(X_0, X_1)$ with $Z_0 \sim \mu_0$ and $Z_1 \sim \mu_1$ by setting $Z_0 = X_0$ and $Z_1$ to the solution of $\dot{Z}_t = v_t(Z_t)$ at time 1.

The authors of [35] prove that this procedure always reduces the transport distance of the coupling, that is, it holds $\mathbb{E}[\|Z_0 - Z_1\|^2] \leq \mathbb{E}[\|X_0 - X_1\|^2]$. Moreover, we can iterate this procedure by generating a sequence $(X_0^{(k+1)}, X_1^{(k+1)}) = \mathcal{R}(X_0^{(k)}, X_1^{(k)})$. Then, the minimal loss function over the first $K$ iteration converges to zero, i.e., it holds $\min_{k=0,\dots,K}\{\min_{w_t} \mathcal{L}(w_t|X_0^{(k)}, X_1^{(k)})\} \to 0$ as $K \to \infty$. Additionally, any coupling $(X_0, X_1)$ with velocity field $v_t$ such that $\mathcal{L}(v_t|X_0, X_1) = 0$ is a fixed point of $\mathcal{R}$. Intuitively, these plans can be characterized by the property that the paths of the ODE (4) are straight. That is, for $\mu_0$-almost every $x$, the solution path $t \mapsto z_t(x)$ of the ODE $\dot{z}_t = v_t(z_t(x))$ has constant velocity in time $v_t(z_t(x))$.

## 2.2   Affine Invariance and Gaussian Case

Next, we consider some equivariances of the rectification step $\mathcal{R}$ with respect to translations and scalings of one or both marginals of the argument $\gamma$. The proofs are given in Appendix A.

**Theorem 2** (Affine Transformations). *Let $(X_0, X_1)$, be a coupling between $\mu_0$ and $\mu_1$, let $v_t = \arg\min_{w_t} \mathcal{L}(w_t|X_0, X_1)$ be the minimizer of the loss function (1) and let $A \in \mathbb{R}^{d \times d}$ be invertible, $b \in \mathbb{R}^d$ and $c \in \mathbb{R}_{>0}$. Then, the following holds true.*

  (i) *The velocity $v_t^{A,b} = \arg\min_{w_t} \mathcal{L}(w_t|AX_0 + b, AX_1 + b)$ is given by $v_t^{A,b}(x) = Av_t(A^{-1}(x-b))$.*

  (ii) *The velocity $v_t^b = \arg\min_{w_t} \mathcal{L}(w_t|X_0, X_1 + b)$ is given by $v_t^b(x) = v_t(x - tb) + b$.*

  (iii) *The velocity $v_t^c = \arg\min_{w_t} \mathcal{L}(w_t|X_0, cX_1)$ is given by*

$$v_t^c = \frac{c}{1-t+ct}v_r\left(\frac{x}{1-t+tc}\right) + \frac{c-1}{1-t+tc}x, \quad with \quad r = \frac{tc}{1-t+tc}.$$

*If $(X_0, X_1)$ is in addition rectifiable with $(Z_0, Z_1) = \mathcal{R}(X_0, X_1)$, then it holds that $\mathcal{R}(AX_0 + b, AX_1 + b) = (AZ_0 + b, AZ_1 + b)$, $\mathcal{R}(X_0, X_1 + b) = (Z_0, Z_1 + b)$ and $\mathcal{R}(X_0, cX_1) = (Z_0, cZ_1)$.*

The invariances (ii) and (iii) hold also true for the optimal transport and the corresponding velocity fields from the Bernamou-Brenier theorem. Part (i) is false for optimal transport and we will see in Remark 7 that it is no longer true if we use the loss function (7) instead of (1). In the specific case that $\mu_0$ and $\mu_1$ are Gaussian and the joint distribution of $(X_0, X_1)$ is a Gaussian as well, we can write down analytically the velocity field $v_t$ which solves (1). Additionally, for the independent coupling of two Gaussians sharing the same eigenvectors, we can show that already the first rectification step leads to the optimal coupling. The proof is given in Appendix A.

**Theorem 3** (Gaussian Case). *Assume that $(X_0, X_1) \sim \mathcal{N}(0, \Sigma)$ with $\Sigma = \begin{pmatrix} \Sigma_0 & \Sigma_{10} \\ \Sigma_{01} & \Sigma_1 \end{pmatrix}$, for positive definite $\Sigma_0$ and $\Sigma_1$. Then, the following holds true.*

  (i) *The minimizer $v_t = \arg\min_{w_t} \mathcal{L}(w_t|X_0, X_1)$ of the loss function (1) is given by*

$$v_t(x) = \frac{1}{1-t}\left(((1-t)\Sigma_{01} + t\Sigma_1)\Sigma_t^{-1} - \mathrm{Id}\right)x, \tag{5}$$

  *where $\Sigma_t = \mathrm{Cov}(X_t) = (1-t)^2\Sigma_0 + (1-t)t(\Sigma_{01} + \Sigma_{10}) + t^2\Sigma_1$.*

  (ii) *Let $\Sigma_{01} = \Sigma_{10} = 0$ and assume that $\Sigma_0$ and $\Sigma_1$ can be jointly diagonalized. Then, $(Z_0, Z_1) := \mathcal{R}(X_0, X_1)$ is the unique optimal coupling between $\mu_0$ and $\mu_1$.*

In the special case where $\Sigma_0 = \mathrm{Id}$, part (ii) was already proven in [44, Prop 4.12]. Note that as a direct consequence of Theorem 2 (i) and the explicit representation of the optimal transport for

Gaussian measures, part (ii) of the previous theorem is no longer true if we skip the assumption that $\Sigma_0$ and $\Sigma_1$ can be jointly diagonalized. The authors of [44] also emphasize the one-dimensional case. However, it is straightforward to see that in the one-dimensional case any rectifiable coupling leads to the optimal transport after one step. For completeness, we formalize the result in the following proposition and include a proof in Appendix A.

**Proposition 4.** *Consider the one-dimensional case and let $(X_0, X_1)$ be a rectifiable coupling between $\mu_0$ and $\mu_1$. Then $(Z_0, Z_1) = \mathcal{R}(X_0, X_1)$ is the optimal coupling between $\mu_0$ and $\mu_1$.*

The explicit representation of $v_t$ in the case where the coupling $(X_0, X_1)$ follows a Gaussian distribution can be generalized to Gaussian mixture models by averaging the vector fields induced by the components, as outlined in the following theorem. The proof is included in Appendix A.

**Theorem 5** (Gaussian Mixture Case). *Assume that $(X_0, X_1) \sim \sum_{k=1}^{K} \pi_k \mathcal{N}(m^k, \Sigma^k)$ with $m^k = \begin{pmatrix} m_0^k \\ m_1^k \end{pmatrix}$ and $\Sigma^k = \begin{pmatrix} \Sigma_0^k & \Sigma_{10}^k \\ \Sigma_{01}^k & \Sigma_1^k \end{pmatrix}$ for positive definite $\Sigma_0^k$ and $\Sigma_1^k$. Write $v_t^k$ the velocity field* (5) *for the covariance matrix $\Sigma^k$ and write $w_t^k(x) = v_t^k(x - tm_1^k - (1-t)m_0^k) + m_1^k - m_0^k$. Then, the minimizer $v_t = \arg\min_{w_t} \mathcal{L}(w_t|X_0, X_1)$ of the loss function* (1) *is given by*

$$v_t(x) = \sum_{k=1}^{K} \alpha^k(x) w_t^k(x), \tag{6}$$

*where $\alpha^k(x) = \frac{\pi_k p_t^k(x)}{\sum_{j=1}^{K} \pi_j p_t^j(x)}$, with $p_t^j$ the Gaussian density of $\mathcal{N}(m_t^j, \Sigma_t^j)$ with $m_t^j = tm_1^j + (1-t)m_0^j$ and $\Sigma_t^j = t^2 \Sigma_1^j + (1-t)^2 \Sigma_0^j + t(1-t)(\Sigma_{10}^j + \Sigma_{01}^j)$.*

Note that the same result holds for degenerated GMMs where $\Sigma_0^k$ and $\Sigma_1^k$ are only positive semi-definite. In the case of generative models, it is classical to assume that $X_0 \sim \mathcal{N}(0, \mathrm{Id})$ and $X_1 \sim \sum_{k=1}^{K} \frac{1}{K} \delta_{m^k}$. If $X_0$ and $X_1$ are independent, then $(X_0, X_1)$ follows a (degenerated) GMM and the velocity field $v_t$ solution is explicit and given by (6) with $w_t^k(x) = \frac{m^k - x}{1-t}$, $m_t^k = tm^k$ and $\Sigma_t^k = (1-t)^2 \Sigma_0$. In these cases, it is evident that our goal is not to compute the velocity field exactly, but rather to rely on its approximation by a neural network, a key element which gives the model its generalization properties.

## 3 Rectified Flows and Optimal Transport

Next, we are interested how rectified flows are related to optimal transport. To this end, we first provide an overview over [34] which relates the optimality of the velocity in rectified flows with the condition that they are a gradient field. Afterwards, we study how this condition effects the solutions of (1).

### 3.1 Backgrounds on Rectified Flows with Gradient Fields

As already observed in [35], a coupling $(X_0, X_1)$ with velocity field $v_t$ such that $\mathcal{L}(v_t|X_0, X_1) = 0$ does not necessarily define an optimal transport. Based on the observation that the optimal velocity field from the Benamou-Brenier theorem [7] always admits a potential [4, Thm. 8.3.1], one of the authors of [35] suggests in [34] to impose the additional constraint that the velocity field $v_t$ from (1) has a potential. More precisely, the loss function (1) is altered to

$$v_t \in \underset{w_t \in L^2(\mu_t)}{\arg\min} \mathcal{L}(w_t|X_0, X_1) \quad \boxed{\text{subject to } w_t = \nabla \varphi_t \text{ for some } \varphi_t \colon \mathbb{R}^d \to \mathbb{R}}. \tag{7}$$

This leads to a rectification step analogously to Definition 1 where the loss (1) is replaced by (7).

**Definition 6.** Let $(X_0, X_1)$ be a coupling between $\mu_0$ and $\mu_1$ and denote by $v_t$ the minimizer of the loss function (7). We denote by $(Z_0, Z_1) = \mathcal{R}_p(X_0, X_1)$ the rectification step with potential, where $Z_0 = X_0$ and where $Z_1$ is the solution of $\dot{Z}_t = v_t(Z_t)$ at time 1.

Note that it is unclear whether the minimum in (7) really exists and that this question is not addressed in [34]. In Proposition 8, we will show that such solutions always exist in a weak form and relate them to the minimal-norm solution of the continuity equation as defined in [4, Thm. 8.3.1].

Now, the author of [34] claims that

$$(X_0, X_1) = \mathcal{R}_p(X_0, X_1) \iff \exists v_t = \nabla \varphi_t : \mathcal{L}(v_t | X_0, X_1) = 0 \iff (X_0, X_1) \text{ is optimal.} \qquad (8)$$

We will see later in Section 4.2 that this result requires several assumptions. In particular, we have to assume that $X_t$ has full support for all $t \in (0,1)$, that the minimizer of (7) is sufficiently smooth and that $(X_0, X_1)$ is rectifiable. While the last two assumptions are stated in [34], the first one is missing and we show that without this assumption the claim (8) is indeed false.

Let us stress that Liu [34] also considers rectified flows for more general cost functions $c(x, y)$ than just the quadratic cost. Considering that our examples already appear for the "simple" case of the quadratic cost, we only consider this case.

## 3.2 Velocity Fields with Potential

In the following, we are interested in the effects of imposing that the velocity field admits a gradient. More precisely, we study, how the solutions of problems (1) and (7) differ. To this end, we first consider how the affine invariances from Theorem 2 change in this case. Afterwards, we prove existence of solutions in (7) in a weak form.

**Remark 7** (Affine Invariances with Potential). It is straightforward to show that parts (ii) and (iii) of Theorem 2 are also true if we replace $\mathcal{R}$ by $\mathcal{R}_p$. To this end, we just have to show that $v_t = \nabla \varphi_t$ implies that there exist $\varphi_t^b$ and $\varphi_t^c$ such that $v_t^b = \nabla \varphi^b$ and $v_t^c = \nabla \varphi^c$. This is fulfilled for

$$\varphi_t^b = \varphi_t(x - tb) + \langle b, x \rangle, \quad \text{and} \quad \varphi_t^c = c \varphi_r \left( \frac{x}{1 - t + tc} \right) + \frac{c - 1}{2(1 - t + tc)} \|x\|^2.$$

However, item (i) is not true for $\mathcal{R}_p$. Indeed, a velocity field on $\mathbb{R}^d$ has a potential if and only if the Jacobian is symmetric, see [18, Thm 6.6.3]. For the specific choice of $v_t^{A,b}$, this is the case if and only if $A^T A J v_t(x) = J v_t(x) A^T A$ for all $x$.

In general, it is not clear whether solutions of (7) exist. However, in the following we prove existence in a weak form. To this end, we consider the space $\mathrm{T}_{\mu_t} := \overline{\{\nabla \varphi : \varphi \in C_c^\infty(\mathbb{R}^d)\}}$, where the closure is taken in $L^2(\mu_t)$. Now, we weaken the constraint in (7), by only assuming that $v_t \in \mathrm{T}_{\mu_t}$. More precisely, we consider the problem

$$v_t \in \underset{w_t \in L^2(\mu_t)}{\arg\min} \ \mathcal{L}(w_t | X_0, X_1) \quad \boxed{\text{subject to } w_t \in \mathrm{T}_{\mu_t}}. \qquad (9)$$

The next proposition shows that the solution of (9) is the limit of a minimizing sequence in the optimization problem from (7). In particular, both solutions coincide whenever the minimizer in (7) exists. To this end, we first show that the solution of (9) is the orthogonal projection onto $T_{\mu_t}$ of the solution of (1). The proof is given in Appendix A.

**Proposition 8.** *Let $v_t$ and $v_t^p$ be the solutions of* (1) *and* (9). *Then, the following holds true.*

  (i) *For any $t \in [0,1]$, we have that $v_t^p = \arg\min_{w_t \in \mathrm{T}_{\mu_t}} \|v_t - w_t\|_{L^2(\mu_t)}$.*

  (ii) *There exist $\varphi^n \in C_c^\infty([0,1] \times \mathbb{R}^d)$ such that $\nabla \varphi^n \to v^p$ in $L^2(dt \otimes \mu_t)$ and $\mathcal{L}(\nabla \varphi_t^n | X_0, X_1) \to \inf_{w_t = \nabla \varphi_t} \mathcal{L}(w_t | X_0, X_1)$.*

  (iii) *The vector field $v_t^p$ is the minimal-norm solution of the continuity equation $\partial_t \mu_t + \mathrm{div}(v_t^p \mu_t) = 0$. That is, it minimizes the norm $\int_0^1 \int \|v_t\|^2 d\mu_t dt$ among all solutions of the continuity equation wrt. $\mu_t$.*

  (iv) *If the minimizer in* (7) *exists, then it coincides with $v_t^p$.*

The minimal-norm velocity field from part (iii) was defined in [4, Thm. 8.3.1], see also Ex. 8.5 in [51]. It minimizes the same objective as the Benamou-Brenier theorem. However, we stress that Benamou-Brenier also optimizes over the path $\mu_t$, which is here fixed as the distribution of $X_t$ such that $v_t^p$ does not directly lead to the optimal transport. Following the proposition, we say that $v_t^p$ is a solution of (7), if it is the limit in $L^2(dt \otimes \mu_t)$ of a minimizing sequence of gradients of potentials $\varphi \in C_c^\infty([0,1] \times \mathbb{R}^d)$. Given the universal approximation theorem, this particularly implies that the solution of (7) can be approximated by the gradient of a sufficiently large neural network.

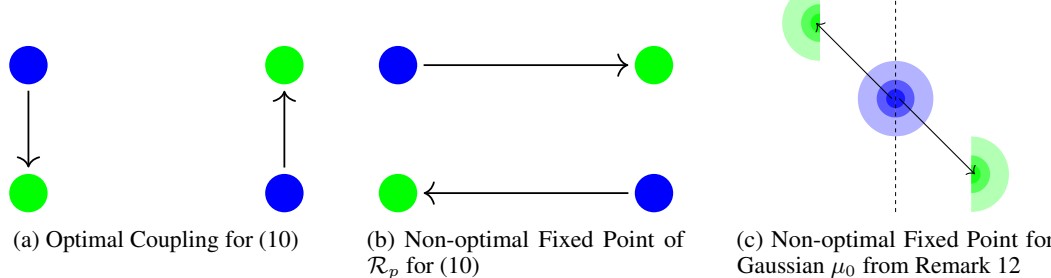

(a) Optimal Coupling for (10)

(b) Non-optimal Fixed Point of $\mathcal{R}_p$ for (10)

(c) Non-optimal Fixed Point for Gaussian $\mu_0$ from Remark 12

Figure 1: Construction of non-optimal couplings which are fixed points of $\mathcal{R}_p$.

If $X_0 \sim \mathcal{N}(0, \mathrm{Id})$ and if $X_0$ and $X_1$ are independent, rectified flows are equivalent to denoising diffusion models as outlined in [35, Section 3.5]. In particular, the solutions of (1) and (7) coincide by the next corollary. The proof is a direct consequence of (2) and Tweedie's formula [16] and can be found, e.g., in [52, Proposition 4.11].

**Corollary 9.** *Let* $(X_0, X_1) \sim \mu_0 \otimes \mu_1$ *with* $\mu_0 = \mathcal{N}(0, \mathrm{Id})$. *Then, it holds that the velocity field* $v_t$ *from* (1) *is given by* $v_t(x) = \frac{1-t}{t} s_t(x) + \frac{1}{t} x$, *where* $s_t(x) = -\nabla \log(p_t(x))$, *is the Stein score for the density* $p_t$ *of* $X_t$. *In particular,* $v_t$ *admits the potential* $v_t = \nabla \varphi_t$ *for* $\varphi_t(x) = -\frac{1-t}{t} \log(p_t(x)) + \frac{1}{2t} \|x\|^2$ *so that the solutions of* (1) *and* (7) *coincide.*

## 4 Counterexamples

In the following, we study specific cases in which the equivalence in (8) is not true. Here, we first consider an example where $\mu_0$ and $\mu_1$ have a disconnected support leading to a fixed point of $\mathcal{R}_p$ which is not the optimal transport plan. Afterwards, we construct a non-rectifiable coupling which has zero loss in (7), but is again not optimal.

### 4.1 Disconnected Supports

We construct a simple example of a non-optimal velocity field which admits straight paths and a potential contradicting (8) and [34, Theorem 5.6]. To this end, let $\eta \in \mathcal{P}(\mathbb{R}^2)$ be an arbitrary probability measure with $\mathrm{support}(\eta_0) \subseteq \{x \in \mathbb{R}^2 : \|x\| \leq 0.3\}$ and denote by $\eta^b = (\cdot + b)_{\#} \eta$ shifted versions of $\eta$. Then, we define

$$\mu_0 = \frac{1}{2} \left( \eta^{(-2,1)} + \eta^{(2,-1)} \right) \quad \text{and} \quad \mu_1 = \frac{1}{2} \left( \eta^{(-2,-1)} + \eta^{(2,1)} \right), \tag{10}$$

see Figure 1a and 1b for an illustration. Now let $X_0 = \tilde{X}_0 \sim \mu_0$ and define

$$X_1 = \begin{cases} X_0 - (0,2) & \text{if } (X_0)_1 < -1 \\ X_0 + (0,2) & \text{if } (X_0)_1 > 1 \end{cases} \quad \text{and} \quad \tilde{X}_1 = \begin{cases} X_0 - (4,0) & \text{if } (X_0)_2 < -0.5 \\ X_0 + (4,0) & \text{if } (X_0)_2 > 0.5 \end{cases}, \tag{11}$$

see Figure 1a and 1b for an illustration. Then the following proposition shows that the coupling $(\tilde{X}_0, \tilde{X}_1)$ is a counterexample to (8). The proof is a straightforward calculation. For completeness, we include it in Appendix B.

**Proposition 10.** *Let* $(X_0, X_1)$ *and* $(\tilde{X}_0, \tilde{X}_1)$ *be defined as above. Then, both couplings* $(X_0, X_1)$ *and* $(\tilde{X}_0, \tilde{X}_1)$ *are fixed points of* $\mathcal{R}_p$ *and have zero loss in* (7), *i.e.,*

$$\min_{w_t = \nabla \varphi_t} \mathcal{L}(w_t | X_0, X_1) = \min_{w_t = \nabla \varphi_t} \mathcal{L}(w_t | \tilde{X}_0, \tilde{X}_1) = 0.$$

*Moreover, it holds* $\mathbb{E}[\|\tilde{X}_1 - \tilde{X}_0\|^2] > \mathbb{E}[\|X_1 - X_0\|^2]$ *such that the coupling* $(\tilde{X}_0, \tilde{X}_1)$ *is not optimal.*

The proof in [34, Theorem 5.6] fails, since in the direction ii)→iii) the author only shows that any velocity field $v_t$ with $v_t = \nabla \varphi_t$ and $\mathcal{L}(v_t | X_0, X_1) = 0$ has straight paths $X_t$-*almost everywhere*. However, they then use [34, Lemma 5.9] which requires that the velocity field has straight paths *everywhere*. Our example shows that this assumption cannot be neglected. The statement of [34,

Theorem 5.6] is correct if we assume that $\mathrm{supp}(X_t) = \mathbb{R}^d$ is connected in addition to the other assumptions of the theorem, although we conjecture that the smoothness assumptions on $\varphi$ (assumed in $C^{2,1}(\mathbb{R}^d \times [0,1])$ in [34, Theorem 5.6]) can probably be lowered. The corrected statement of [34, Theorem 5.6] reads as follows. The proof is the same as in [34].

**Theorem 11.** *Assume that $(X_0, X_1)$ is rectifiable and let $v_t = \nabla \varphi_t \in \mathrm{argmin}_{w_t = \nabla \psi_t} \mathcal{L}(w_t | X_0, X_1)$ fulfill that $\varphi_t \in C^{2,1}(\mathbb{R}^d \times [0,1])$. Moreover, suppose that $\mathrm{supp}(X_t) = \mathbb{R}^d$ for $X_t = (1-t)X_0 + tX_1$. Then it holds*

$$\mathcal{R}_p(X_0, X_1) = (X_0, X_1) \quad \Leftrightarrow \quad \mathcal{L}(v_t | X_0, X_1) = 0 \quad \Leftrightarrow \quad (X_0, X_1) \text{ is an optimal coupling.}$$

In [34, Section 6], the author raises the question, whether for $(Z_0^{(i+1)}, Z_1^{(i+1)}) = \mathcal{R}_p(Z_0^{(i)}, Z_1^{(i)})$ the optimality gap

$$\mathbb{E}[\|Z_1^{(i)} - Z_0^{(i)}\|^2] - \inf_{Z_0 \sim \mu_0, Z_1 \sim \mu_1} \mathbb{E}[\|Z_1 - Z_0\|^2]$$

converges to zero for $i \to \infty$. The above example gives a negative answer to this question, since it leads to a constant but strictly positive optimality gap.

**Remark 12.** Many applications consider the case where $\mu_0$ is a standard normal distribution. However, we note that we can construct a similar counterexample for this case. To this end let $X_0 \sim \mathcal{N}(0, \mathrm{Id})$ and define

$$X_1 = \begin{cases} X_0 + (-2, 2) & \text{if } (X_0)_1 < 0, \\ X_0 - (-2, 2) & \text{if } (X_0)_1 > 0, \end{cases} \tag{12}$$

see Figure 1c for an illustration. With similar arguments as for the previous example, we can observe that $(X_0, X_1)$ is a fixed point of $\mathcal{R}_p$ but is not optimal. The full statement and a corresponding proof are included as Proposition 19 in Appendix B. Note that also in this example, the support of $X_t$ is disconnected for any $t > 0$. It is also possible to make a more complicated counterexample where the $X_t$ have full support but with very irregular transport and potential, see Example 20 in Appendix B.

We verify numerically that the couplings discussed in this subsection are indeed fixed points of $\mathcal{R}_p$ in Appendix D.

## 4.2 Non-Rectifiable Couplings

In this subsection, we consider the case of non-rectifiable couplings in more detail. More precisely, we give an example of a non-rectifiable coupling $(X_0, X_1)$ such that the minimizer $v_t = \nabla \varphi_t$ of (7) fulfills $\mathcal{L}(v_t | X_0, X_1) = 0$, showing that (8) is again false in this case. Afterwards, we provide some sufficient condition for a coupling $\gamma$ to be rectifiable.

We consider $\mu_0 = \mu_1 = \mathcal{N}(0, \mathrm{Id})$ and define the coupling $(X_0, X_1)$ by $X_1 = -X_0$, which is illustrated in Figure 2. Then, the next proposition shows that the loss from (7) is indeed zero and that $(X_0, X_1)$ is nonoptimal. The proof is given in Appendix B.

**Proposition 13.** *Let $(X_0, X_1)$ be defined as above. Then, the following holds true.*

(i) *The minimizer $v_t = \arg\min_{w_t \in L^2(\mu_t)} \mathcal{L}(v_t | X_0, X_1)$ is given by $v_t(x) = -\frac{2}{1-2t}x = \nabla \varphi_t(x)$ for $\varphi_t(x) = -\frac{1}{1-2t}\|x\|^2$ for $t \neq \frac{1}{2}$ and by $v_t(x) = 0$ for $t = \frac{1}{2}$.*

(ii) *It holds $\mathcal{L}(v_t | X_0, X_1) = 0$ even though $(X_0, X_1)$ is not optimal.*

(iii) *The coupling $(X_0, X_1)$ is not rectifiable, i.e., the ODE $\dot{Z}_t = v_t(Z_t)$ does not admit a unique solution.*

The next theorem provides a sufficient condition which ensures that a given coupling is rectifiable. The proof is given in Appendix B.

**Theorem 14.** *Let $(X_0, X_1)$ be a coupling between $\mu_0$ and $\mu_1$ and denote by $P_{X_0 | X_1 = x_1}$ the conditional distribution of $X_0$ given $X_1 = x_1$. If $P_{X_0 | X_1 = x_1}$ is absolutely continuous with a smooth and positive density $p_{X_0 | X_1 = x_1}(x_0)$, then $(X_0, X_1)$ is rectifiable and the solutions $v_t$ and $v_t^p$ of (1) and (7) are smooth.*

We highlight two examples, where the assumptions of the theorem are fulfilled.

**Example 15.** Let $\mu_0$ be absolutely continuous with smooth density and consider the independent coupling $(X_0, X_1) = \mu_0 \otimes \mu_1$. Then, the conditional distribution $P_{X_0|X_1=x_1} = \mu_0$ has a smooth density. In particular, the assumptions of Theorem 14 are fulfilled and $(X_0, X_1)$ is rectifiable.

Nevertheless, since it is not clear that $\mathcal{R}(X_0, X_1)$ is rectifiable when $(X_0, X_1)$ is rectifiable it is still open whether any sequence generated by $(X_0^{(i+1)}, X_1^{(i+1)}) \in \mathcal{R}(X_0^{(i)}, X_1^{(i)})$ remains rectifiable if the initial coupling $(X_0^{(0)}, X_1^{(0)})$ is rectifiable. The next example shows that any coupling can be made rectifiable by injecting an arbitrary small amount of noise.

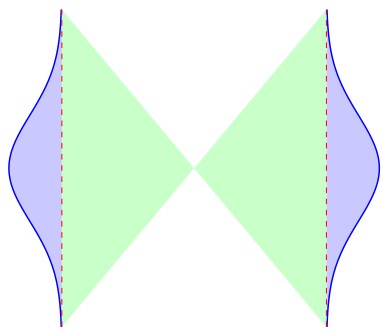

Figure 2: Illustration of the non-rectifiable coupling $(X_0, X_1)$ with $X_0 = -X_1 \sim \mathcal{N}(0, \text{Id})$

**Example 16.** For any coupling $(X_0, X_1)$ between $\mu_0$ and $\mu_1$ and an independent noise variable $W \sim \mathcal{N}(0, \text{Id})$, we can define a smoothed coupling $(X_0^c, X_1^c) := (X_0 + cW, X_1)$ between $\mu_0^c := \mu_0 * \mathcal{N}(0, c^2 \text{Id})$ and $\mu_1$. By Theorem 14 this coupling is guaranteed to be rectifiable. Note that this procedure alters the left marginal $\mu_0$ by the convolution with a Gaussian. But if $c$ becomes small, it becomes arbitrarily close to $\mu_0$.

As a consequence of Proposition 13 and Example 16, we obtain that a small loss value $\mathcal{L}(v_t|X_0, X_1)$ for $v_t = \arg \min_{w_t = \nabla \varphi_t} \mathcal{L}(w_t|X_0, X_1)$ does not imply that the velocity field of $v_t$ and coupling $(X_0, X_1)$ are close to be optimal. We formalize this finding in the following corollary. The proof is given in Appendix B.

**Corollary 17.** Let $(X_0, X_1)$ with $X_0 = -X_1 \sim \mathcal{N}(0, \text{Id})$ be a coupling between $\mu_0 = \mu_1 = \mathcal{N}(0, \text{Id})$. Denote by $(X_0^c, X_1^c)$ the smoothed coupling from Example 16 for $c > 0$ and by $v_t^c = \arg \min_{w_t = \nabla \varphi_t} \mathcal{L}(w_t|X_0^c, X_1^c)$ the corresponding velocity field. Then, for any $\epsilon > 0$, there exists $c > 0$ small enough such that $\mathcal{L}(v_t^c|X_0^c, X_1^c) < \epsilon$ and $W_2^2(\mu_0^c, \mu_1) < \epsilon$, but

$$\int_0^1 \mathbb{E}[\|v_t^c(X_t^c)\|^2]dt > 4 - \epsilon \quad \text{and} \quad \mathbb{E}[\|X_1^c - X_0^c\|^2] > 4 - \epsilon.$$

## 5 Smoothed Rectification

The rectification process proposed by Liu in [34] never ensures that the iterates remain rectifiable, although we understand from the previous section that this property can be crucial. In the following, we propose a smoothing procedure of rectified couplings in order to ensure that the iterates always remain rectifiable. To this end, we require that $\mu_0 \sim \mathcal{N}(0, \text{Id})$. More precisely, starting with some initial coupling $(Z_0^{(0)}, Z_1^{(0)})$ we define a sequence of couplings $(Z_0^{(i)}, Z_1^{(i)})$ by defining $(Z_0^{(i+1)}, Z_1^{(i+1)}) = \mathcal{R}(X_0^{(i)}, X_1^{(i)})$, where $X_0^{(i)} = \sqrt{1 - c_i} Z_0^{(i)} + \sqrt{c_i} W^{(i)}$ with $W^{(i)} \sim \mathcal{N}(0, \text{Id})$ independent of $(Z_0^{(i)}, Z_1^{(i)})$ and $X_1^{(i)} = Z_1^{(i)}$ and some $c_i \in (0, 1)$. The following theorem proves that the iteration still optimizes the loss function up to some error which depends on the injected noise levels $c_i$. The proof is given in Appendix C.

**Theorem 18.** Let $(Z_0^{(i)}, Z_1^{(i)})$ and $(X_0^{(i)}, X_1^{(i)})$ be defined as above. Denote by $L^{(i)} := \inf_{w_t} \mathcal{L}(w_t|X_0^{(i)}, X_1^{(i)})$ the loss values in the rectification steps and by $V_1 = \int \|x\|^2 \, d\mu_1(x)$ the second moment of $\mu_1$. Then, the following holds true.

(i) We have that $(X_0^{(i)}, X_1^{(i)})$ is rectifiable and that $(X_0^{(i)}, X_1^{(i)})$ and $(Z_0^{(i)}, Z_1^{(i)})$ are couplings between $\mu_0$ and $\mu_1$ for all $i$, i.e., that $X_0^{(i)}, Z_0^{(i)} \sim \mu_0$ and $X_1^{(i)}, Z_1^{(i)} \sim \mu_1$;

(ii) For $\bar{c}_K = \frac{1}{K} \sum_{i=0}^{K-1} c_i$, it holds $\min_{i=0,\dots,K-1} L^{(i)} \in \mathcal{O}\left(\frac{1}{K} + \bar{c}_K\right)$.

For constant noise levels $c_i = c$, Theorem 18 states that the minimal loss value of the iterates tends to zero up to an error which depends linearly on the variance $c$ of the injected noise. However, as soon

as the noise levels tend to zero, also the averages $\bar{c}_K$ converge to zero. In particular, for summable noise levels with $\sum_{i=1}^{\infty} c_i = C < \infty$, we have that $\min_{i=0,\ldots,K-1} L_k \in \mathcal{O}(1/K)$, which is the same rate as in [34] without noise injection. By the same proof, Theorem 18 also holds true if we replace the step $(Z_0^{(i+1)}, Z_1^{(i+1)}) = \mathcal{R}(X_0^{(i)}, X_1^{(i)})$ by $(Z_0^{(i+1)}, Z_1^{(i+1)}) = \mathcal{R}_p(X_0^{(i)}, X_1^{(i)})$.

Additionally, we note that we always have $\mathrm{supp}(X_t) = \mathbb{R}^d$ within the smoothed rectification. Therefore, also the counterexamples from Section 4.1 do not apply in this case. However, it remains still unclear whether the smoothed rectification converges to optimal transport. Numerically, we test the approach in Appendix E on the example from Remark 12, where convergence to the optimal transport indeed seems to be fulfilled.

# 6 Conclusions

Even though rectified flows have shown to define efficient generative models in the literature, we have seen in this paper that they are not a suitable tool for computing optimal transport maps between two distributions. In particular, we have identified the following two main reasons for that:

- **Non-optimal fixed points of $\mathcal{R}_p$:** In the case where the distributions $\mu_t$ have disconnected support, we showed in Section 4.1 that there exist fixed points of the rectification step, where the resulting velocity field has a potential but does not lead to the optimal coupling. In particular, [34, Theorem 5.6] is not true without additional assumptions. Given that datasets in applications are often disconnected this limits the applicability of rectified flows for computing optimal couplings significantly.

- **Vanishing loss does not imply optimality:** All convergence guarantees in [34, 35] state that the loss function (1) or (7) becomes arbitrary small. However, already in the simple case that $(X_0, X_1)$ follows a Gaussian distribution, we showed in Corollary 17 that there exist couplings with an arbitrary small loss function which are arbitrary far away from the optimal coupling.

Moreover, we studied the assumption that a coupling is rectifiable. While we can indeed give a simple example where this assumption is violated, they are heavily based on symmetry such that they probably will not appear in practice. Indeed, we show that injecting a small amount of noise in each step ensures that couplings remain rectifiable and do not alter the theoretical guarantees.

**Limitations and Open Questions** While we have shown the existence of non-optimal fixed points of $\mathcal{R}_p$ it is unclear to us, to which fixed point the iterative rectification converges. This question will heavily depend on the initial coupling. Here, interesting cases to consider would be the independent coupling and couplings defined by minibatch optimal transport [43, 49]. Additionally, a noisy version of rectified flows was considered in [1, 14, 46] in connection with Schrödinger bridges. While in this case convergence to the entropic optimal transport plan is guaranteed, we show in Appendix F numerically that this convergence can become arbitrary slow for small regularization parameters. Similar as for rectified flowsm it could be interesting to consider the convergence behavior locally around the (regularized) optimal transport plan.

The same limitation applies to non-rectifiable couplings. Even though we showed the existence of non-rectifiable couplings, we do not know so far whether they appear during the iterative rectification when we start with a "smooth" coupling like the independent one. Moreover, the noise injection from Section 5 guaranteeing the rectifiability throughout the iterations is so far limited to the case where $\mu_0$ is Gaussian. While we can show the same convergence result for the rectification with and without noise-injection, these result only states that the loss converges to zero. However, we have seen in Section 4.1 and Corollary 17 that this is not sufficient to show convergence to the optimal coupling or even to a fixed point of $\mathcal{R}_p$.

Finally, we restricted our considerations to the case of the quadratic cost function, while [34] considers more general choices. However, the counterexamples from Section 4.2 are independent of the cost function as the issue arises from the fact that the interpolation $X_t$ degenerates. Also the counterexamples from Section 4.1 can be transferred to more general cost functions with very similar arguments.

## Acknowledgements

JH is funded by the Deutsche Forschungsgemeinschaft (DFG, German Research Foundation) within project no 530824055. AC and JD acknowledge the support of the "France 2030" funding ANR-23-PEIA-0004 ("PDE-AI"). AC also acknowledges the support of the "France 2030" funding ANR-23-IACL-0008 ("PR[AI]RIE-PSAI").

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

# A  Proofs from Section 2 and 3

*Proof of Theorem 2.* For the proof, we use the representation of $v_t$ via the conditional expectation from (2). If $(X_0, X_1)$ is rectifiable, we use the notation $Z_t$ for the random variable defined as $Z_0 = X_0$ and $\dot{Z}_t = v_t(Z_t)$.

(i) By definition it holds that

$$v_t^{A,b}(x) = \mathbb{E}[AX_1 + b - (AX_0 + b)|AX_t + b = x] = \mathbb{E}[A(X_1 - X_0)|AX_t + b = x]$$
$$= A\mathbb{E}[X_1 - X_0|X_t = A^{-1}(x - b)] = Av_t(A^{-1}(x - b)).$$

If $(X_0, X_1)$ is rectifiable, we observe for $Z_t^{A,b} = AZ_t + b$ that $Z_0^{A,b} = AZ_0 + b = AX_0 + b = Z_0$ and

$$\dot{Z}_t^{A,b}(x) = A\dot{Z}_t + b = Av_t(Z_t) + b = Av_t(A^{-1}(Z_t^{A,b} - b)) = v_t^{A,b}(Z_t^{A,b}).$$

Thus, we have $(AZ_0 + b, AZ_1 + b) = \mathcal{R}(AX_0 + b, AX_1 + b)$.

(ii) We have by definition that

$$v_t^b(x) = \mathbb{E}[X_1 + b - X_0|X_t + tb = x]$$
$$= \mathbb{E}[X_1 - X_0|X_t = x - tb] + b = v_t(x - tb) + b.$$

If $(X_0, X_1)$ is rectifiable, we observe for $Z_t^b = Z_t + tb$ that $Z_0^b = Z_0 = X_0$ and

$$\dot{Z}_t^b(x) = \dot{Z}_t + b = Av_t(Z_t) + b = v_t(Z_t^b - tb) + b = v_t^b(Z_t^b).$$

Thus, we have $(Z_0, Z_1 + b) = \mathcal{R}(X_0, X_1 + b)$.

(iii) Denote $(X_0^c, X_1^c) = (X_0, cX_1)$. Then, it holds

$$X_t^c := (1 - t)X_0^c + tX_1^c = (1 - t)X_0 + tcX_1$$
$$= (1 - t + tc)\left(\frac{1 - t}{1 - t + tc}X_0 + \frac{tc}{1 - t + tc}X_1\right) = (1 - t + tc)X_r$$

with $r = \frac{tc}{1 - t + tc}$. Thus, we have

$$v_t^c(x) = \frac{1}{1 - t}\left(\mathbb{E}[X_1^c|X_t^c = x] - x\right) = \frac{1}{1 - t}\left(\mathbb{E}[cX_1|(1 - t + tc)X_r = x] - x\right)$$
$$= \frac{1}{1 - t}\left(c\mathbb{E}\left[X_1|X_r = \frac{x}{1 - t + tc}\right] - x\right)$$
$$= \frac{c}{1 - t}\left(\mathbb{E}\left[X_1|X_r = \frac{x}{1 - t + tc}\right] - \frac{x}{1 - t + tc}\right) + \frac{c - 1}{1 - t + tc}x$$
$$= \frac{c}{1 - t + tc}v_r\left(\frac{x}{1 - t + tc}\right) + \frac{c - 1}{1 - t + tc}x$$

If $(X_0, X_1)$ is rectifiable, we observe for $Z_t^c = (1 - t + tc)Z_r$ that

$$\dot{Z}_t^c = (1 - t + tc)\dot{r}\dot{Z}_r + (c - 1)Z_r = (1 - t + tc)\dot{r}v_r(Z_r) + (c - 1)Z_r$$
$$= \frac{c}{1 - t + tc}v_r\left(\frac{Z_t^c}{1 - t + tc}\right) + \frac{c - 1}{1 - t + tc}Z_t^c = v_t^c(Z_t^c),$$

where we used $\dot{r} = \frac{c}{(1 - t + tc)^2}$. Thus, we have $(Z_0, cZ_1) = \mathcal{R}(X_0, cX_1)$.

$\square$

*Proof of Theorem 3.* We obtain by the calculation rules for Gaussian distributions that $(X_t, X_1) \sim \mathcal{N}(0, \Sigma^t)$ with

$$\Sigma^t = \begin{pmatrix} \Sigma_t & \Sigma_{1t} \\ \Sigma_{t1} & \Sigma_1 \end{pmatrix} := \begin{pmatrix} (1-t)\mathrm{Id} & t\mathrm{Id} \\ 0 & \mathrm{Id} \end{pmatrix} \begin{pmatrix} \Sigma_0 & \Sigma_{10} \\ \Sigma_{01} & \Sigma_1 \end{pmatrix} \begin{pmatrix} (1-t)\mathrm{Id} & 0 \\ t\mathrm{Id} & \mathrm{Id} \end{pmatrix}$$

$$= \begin{pmatrix} (1-t)^2\Sigma_0 + (1-t)t(\Sigma_{01} + \Sigma_{10}) + t^2\Sigma_1 & (1-t)\Sigma_{10} + t\Sigma_1 \\ (1-t)\Sigma_{01} + t\Sigma_1 & \Sigma_1 \end{pmatrix}.$$

Thus, we have that $\mathbb{E}[X_1|X_t = x] = \Sigma_{t1}\Sigma_t^{-1}x$. Inserting this formula in $v_t(x) = \frac{1}{1-t}\left(\mathbb{E}[X_1|X_t = x] - x\right)$ proves the first part.

For part (ii), note that we will see later in Example 15 that $(X_0, X_1)$ is rectifiable. Then, the claim was proven for $\Sigma_0 = \mathrm{Id}$ in [44, Proposition 4.12]. The general case follows from Theorem 2 (i). $\square$

*Proof of Proposition 4.* Let $v_t = \arg\min_{w_t} \mathcal{L}(w_t|X_0, X_1)$ and define by $z_t(x)$ the solution of the ODE $\dot{z}_t(x) = v_t(z_t(x))$ with initial condition $z_0(x) = x$. By Brenier's theorem [9], it suffices to show that $z_1 \colon \mathbb{R} \to \mathbb{R}$ has a convex potential, which is equivalent to $z_1$ being monotone increasing. Assume in contrary that there exist $\tilde{x} < x$ such that $z_1(\tilde{x}) > z_1(x)$. Since the mapping $f(t) := z_t(\tilde{x}) - z_t(x)$ is continuous and $f(0) = \tilde{x} - x < 0 < z_1(\tilde{x}) - z_1(x) = f(1)$, there exists some $t \in (0, 1)$ such that $f(t) = 0$. This implies that $z_t(\tilde{x}) = z_t(x)$ which means that the ODE $\dot{z}_t(x) = v_t(z_t(x))$ admits crossing paths and contradicts the uniqueness of the solution. $\square$

*Proof of Theorem 5.* Writing $Z$ the latent variable for the mixture, we can always write $\mathbb{E}[X_1|X_t = x] = \mathbb{E}[\mathbb{E}[X_1|X_t = x, Z]|X_t = x] = \sum_{k=1}^K \mathbb{P}[Z = k|X_t = x]\mathbb{E}[X_1|X_t = x, Z = k]$. Using $v_t(x) = \frac{1}{1-t}\left(\mathbb{E}[X_1|X_t = x] - x\right)$, it follows that

$$v_t(x) = \sum_{k=1}^K \mathbb{P}[Z = k|X_t = x]v_t^k(x) = \sum_{k=1}^K \alpha^k(x)v_t^k(x).$$

$\square$

*Proof of Proposition 8.* For part (i), let $w_t \in \mathrm{T}_{\mu_t}$ and $v_t$ be the solution of (1). Then we have that

$$\mathcal{L}(w_t|X_0, X_1) = \int_0^1 \mathbb{E}[\|w_t(X_t) - X_1 + X_0\|^2]\, dt$$

$$= \int_0^1 \mathbb{E}[\|w_t(X_t) - v_t(X_t) + v_t(X_t) - X_1 + X_0\|^2]\, dt$$

$$= \int_0^1 \mathbb{E}[\|w_t(X_t) - v_t(X_t)\|^2]\, dt + \int_0^1 \mathbb{E}[\|v_t(X_t) - X_1 + X_0\|^2]\, dt$$

$$+ 2\int_0^1 \mathbb{E}[\langle w_t(X_t) - v_t(X_t), v_t(X_t) - X_1 + X_0\rangle]\, dt$$

It remains to show that the last term is zero. To this end, we note that

$$\mathbb{E}[\langle w_t(X_t) - v_t(X_t), v_t(X_t) - X_1 + X_0\rangle]$$
$$= \mathbb{E}\left[\mathbb{E}[\langle w_t(X_t) - v_t(X_t), v_t(X_t) - X_1 + X_0\rangle|X_t]\right]$$
$$= \mathbb{E}\left[\langle w_t(X_t) - v_t(X_t), \mathbb{E}[v_t(X_t) - X_1 + X_0|X_t]\rangle\right]$$
$$= \mathbb{E}\left[\langle w_t(X_t) - v_t(X_t), v_t(X_t) - \mathbb{E}[X_1 - X_0|X_t]\rangle\right] = 0,$$

where the first step comes from the properties of conditional expectations, the second and third step uses that $w_t(X_t)$ and $v_t(X_t)$ are $X_t$-measurable and the last step uses that $v_t(X_t) = \mathbb{E}[X_1 - X_0|X_t]$.

For part (ii) and (iii), we prove here that one can define, in a weak form, solutions to (7). We introduce the measure $dt \otimes \mu_t$, defined by

$$\int \varphi(t, x)dt \otimes \mu_t = \int_0^1 \int \varphi(t, x)d\mu_t dt$$

for any test function $\varphi \in C_c([0, 1] \times \mathbb{R}^d)$, and the space $L^2(dt \otimes \mu_t) \simeq L^2([0, 1]; L^2(\mu_t))$. Consider now a minimizing sequence for (7), of the form $(\nabla \varphi^n)_{n \geq 1}$ where each function $\varphi^n \in C_c^\infty([0, 1] \times \mathbb{R}^d)$. Thanks to part (i), we know that $(\nabla \varphi^n)_{n \geq 1}$ is also a minimizing sequence for

$$\inf \left\{ \int_0^1 \int_{\mathbb{R}^d} \|\nabla \varphi_t - v_t\|^2 d\mu_t dt : \varphi \in C_c^\infty([0, 1] \times \mathbb{R}^d) \right\} \tag{13}$$

where $v_t$ is the solution of (1), which is such that (3) holds (in the distributional sense). Up to a subsequence, $\nabla \varphi^n$ has a weak limit $w$ in $L^2(dt \otimes \mu_t)$, which is such that

$$\int_0^1 \int_{\mathbb{R}^d} \|w_t - v_t\|^2 d\mu_t dt \leq \liminf_n \int_0^1 \int_{\mathbb{R}^d} \|\nabla \varphi_t^n - v_t\|^2 d\mu_t dt.$$

Now, thanks to Mazur's lemma (see, e.g., the text book [3, Lem 8.14]), $w$ is also the strong limit of convex combinations of the $\nabla \varphi^n$: there exist $\theta_{n,m} \in [0, 1]$ such that for each $n$, $\sum_m \theta_{n,m} = 1$ and all $\theta_{n,m}$ vanish but a finite number, and defining $\tilde{\varphi}^n := \sum_m \theta_{n,m} \varphi^m$ one has that $\nabla \tilde{\varphi}^n \to w$ strongly in $L^2(dt \otimes \mu_t)$. Obviously, $(\nabla \tilde{\varphi}^n)_{n \geq 1}$ is also a minimizing sequence for (7), and by construction, one has:

$$\int_0^1 \int_{\mathbb{R}^d} \|w_t - v_t\|^2 d\mu_t dt = \lim_n \int_0^1 \int_{\mathbb{R}^d} \|\nabla \tilde{\varphi}_t^n - v_t\|^2 d\mu_t dt = \text{(value of (13))}$$

Let now $\psi \in C_c^\infty([0, 1] \times \mathbb{R}^d)$ and $s \in \mathbb{R}$, $\|s\| \ll 1$: one has

$$\int_0^1 \int_{\mathbb{R}^d} \|w_t + s\nabla \psi_t - v_t\|^2 d\mu_t dt = \lim_n \int_0^1 \int_{\mathbb{R}^d} \|\nabla(\tilde{\varphi}^n + s\psi)_t - v_t\|^2 d\mu_t dt \geq \text{(value of (13))}.$$

Differentiating at $s = 0$ we deduce that for any test function $\psi$,

$$\int_0^1 \int_{\mathbb{R}^d} \nabla \psi \cdot (w_t - v_t) \mu_t dt = 0$$

which precisely means that in the distributional sense,

$$\text{div}(w_t \mu_t) = \text{div}(v_t \mu_t),$$

showing that the continuity equation (3) holds with the speed $v_t$ replaced with $w_t$. In addition, for a.e. $t \in [0, 1]$, by construction, $w_t$ belongs to $\mathrm{T}_{\mu_t}$, so that by uniqueness it is the vector field with minimal norm defined in [4, Thm. 8.3.1] (see also equation (8.0.1)). Observe that without any knowledge on the regularity of $w$, it is unclear how to associate a unique transport map $X(1, x)$ satisfying $\dot{X} = w(X)$, $X(0, x) = x$ for all $x$.

Part (iv) follows from the facts that any $w_t = \nabla \varphi \in L^2(\mu_t)$ belongs to $\mathrm{T}_{\mu_t}$ and that $v_t^p$ is the limit of a minimizing sequence in (7). $\qquad\square$

## B  Proofs from Section 4

*Proof of Proposition 10.* Define the potentials $\varphi_t$ and $\tilde{\varphi}_t$ such that

$$\varphi_t(x) = \begin{cases} \langle x, (0, -2) \rangle, & \text{if } x_1 < -1, \\ \langle x, (0, 2) \rangle, & \text{if } x_1 > 1, \end{cases} \quad \text{and} \quad \tilde{\varphi}_t(x) = \begin{cases} \langle x, (-4, 0) \rangle, & \text{if } x_2 < -0.5, \\ \langle x, (4, 0) \rangle, & \text{if } x_2 > 0.5, \end{cases}$$

and extend them smoothly to the full $\mathbb{R}^2$. Then, the velocity fields $v_t = \nabla \varphi_t$ and $\tilde{v}_t = \nabla \tilde{\varphi}_t$ are given by

$$v_t(x) = \begin{cases} (0, -2), & \text{if } x_1 < -1, \\ (0, 2), & \text{if } x_1 > 1, \end{cases} \quad \text{and} \quad \tilde{v}_t(x) \begin{cases} (-4, 0), & \text{if } x_2 < -0.5, \\ (4, 0), & \text{if } x_2 > 0.5, \end{cases}$$

Plugging in the definitions of $(X_0, X_1)$ and $(\tilde{X}_0, \tilde{X}_1)$ this yields that

$$\mathcal{L}(v_t | X_0, X_1) = \mathcal{L}(\tilde{v}_t | \tilde{X}_0, \tilde{X}_1) = 0$$

and that $\dot{X}_t = v_t(X_t)$ and $\dot{\tilde{X}}_t = \tilde{v}_t(\tilde{X}_t)$ such that $(X_0, X_1)$ and $(\tilde{X}_0, \tilde{X}_1)$ are fixed points of $\mathcal{R}_p$. On the other side, we directly obtain that

$$\mathbb{E}[\|\tilde{X}_1 - \tilde{X}_0\|^2] = 16 > 4 = \mathbb{E}[\|X_1 - X_0\|^2].$$

$\qquad\square$

**Proposition 19** (Formal Statement of Remark 12). *Let $d = 2$, $X_0 \sim \mathcal{N}(0, \mathrm{Id})$ and define*

$$X_1 = \begin{cases} X_0 + (-2, 2) & \text{if } (X_0)_1 < 0, \\ X_0 - (-2, 2) & \text{if } (X_0)_1 > 0. \end{cases}$$

*Then it holds $(X_0, X_1) = \mathcal{R}_p(X_0, X_1)$, but $(X_0, X_1)$ is not an optimal coupling.*

*Proof.* For $t > 0$ we define the potentials

$$\varphi_t(x) = \begin{cases} \langle x, (-2, 2) \rangle & \text{if } x_1 < -t, \\ \langle x, (2, -2) \rangle & \text{if } x_1 > t, \end{cases}$$

and extend them smoothly to the full $\mathbb{R}^d$. Then, the velocity fields $v_t = \nabla \varphi_t$ are given by

$$v_t(x) = \begin{cases} (-2, 2) & \text{if } x_1 < -t, \\ (2, -2) & \text{if } x_1 > t. \end{cases}$$

Noting that $(X_0, X_t, X_1)$ is given by plugging in the definition of $(X_0, X_1)$ this yields $\mathcal{L}(v_t | X_0, X_1) = 0$ and $\dot{X}_t = v_t(X_t)$ such that $(X_0, X_1) = \mathcal{R}_p(X_0, X_1)$.

Next, we show that $(X_0, X_1)$ is not optimal by contradiction. To this end, note that $X_1 = \mathcal{T}(X_0)$ where $\mathcal{T}(x)$ is defined for $x = (x_1, x_2)$ as

$$\mathcal{T}(x) = \begin{cases} x + (-2, 2) & \text{if } x_1 < 0, \\ x - (-2, 2) & \text{if } x_1 > 0. \end{cases}$$

Assuming that $(X_0, X_1)$ is optimal, there exists by Brenier's theorem [9] some convex function $\psi \colon \mathbb{R}^2 \to \mathbb{R}$ such that $\mathcal{T} = \nabla \psi$ almost everywhere.

Then, we should have $\psi(x) = \psi^+(x) := \|x\|^2/2 + \langle (-2, 2), x \rangle + c^-$ for $x_1 < 0$ and $\psi(x) = \psi^-(x) := \|x\|^2/2 - \langle (-2, 2), x \rangle + c^+$ for $x_1 > 0$, for some constants $c^+, c^- \in \mathbb{R}$. Yet if we want $\psi$ to be continuous across the interface $\{x_1 = 0\}$, we need that $\psi^+(x) = \psi^-(x)$ there, which boils down to $\langle (-2, 2), x \rangle = \text{constant}$: this means that the interface should be normal to $(-2, 2)$, which clearly is not the case. $\square$

**Example 20** (Counterexample regularity/support). We show here that it is also possible to find a path $(\mu_t)_{t \in [0,1]}$ of measures with *full support* at all time, which are fixed point for the rectification. However, the corresponding potential are not regular (or rather, do not really exist) and we do not expect the existence of a transport map in this case. Let $(x_n)_{n \geq 0}$ be such that $\mathbb{Q}^2 = \{x_n : n \in \mathbb{N}\}$, and such that both $\{x_n : n \in 2\mathbb{N}\}$ and $\{x_n : n \in 2\mathbb{N} + 1\}$ are dense in $\mathbb{R}^2$. Let then $(a_n)_{n \geq 0}$ a sequence of positive numbers with $\sum_n a_n = 1$, $e_0, e_1$ two vectors (specified later), and define:

$$\mu_0 = \sum_{n=0}^{+\infty} a_n \delta_{x_n}, \quad \mu_1 = \sum_{n \in 2\mathbb{N}} a_n \delta_{x_n + e_0} + \sum_{n \in 2\mathbb{N}+1} a_n \delta_{x_n + e_1}.$$

By a slight abuse of notation we denote $n[2]$ the remainder $n \mod 2$, and we see that the straight trajectory between $\mu_0$ and $\mu_1$ is simply

$$\mu_t = \sum_{n=0}^{+\infty} a_n \delta_{x_n + t e_{n[2]}}, \quad t \in [0, 1],$$

which satisfies the continuity equation with the speed $v_t(x) = e_{n[2]}$ if $x = x_n + t e_{n[2]}$, $n \geq 0$. Actually, this is true only if the paths do not cross, that is,

$$\forall n \in 2\mathbb{N}, \forall m \in 2\mathbb{N} + 1, \forall t \in (0, 1), \quad x_m - x_n \neq t(e_0 - e_1).$$

Choosing $e_0, e_1$ such that $e_0 - e_1 = (1, \xi)$ with $\xi \notin \mathbb{Q}$ clearly ensures this property, since if $x_m - x_n = t(e_0 - e_1)$, the first coordinate imposes that $t \in \mathbb{Q}$ and then the second that $\xi \in \mathbb{Q}$, a contradiction.

Since for all $t \in [0, 1]$, $(x_n + t e_{n[2]})_{n \geq 0}$ is dense in $\mathbb{R}^2$, the support (defined classically as the complement of the largest open negligible set) of $\mu_t$ is $\mathbb{R}^2$. Then, solving (1) will return the same

speed. In addition, we can check that $v_t \in T_{\mu_t}$ at all time, so that solving (7) will also return the same speed and $\mu_t$ is a fixed point of the rectification process.

To see that $v_t \in T_{\mu_t}$ (for a given time $t \in [0,1]$), we observe that for fixed $N$, one can find small radii $\rho_n^N$, $n = 0, \dots, N$, such that the balls $B(x_n + t e_{n[2]}, \rho_n^N)$ are disjoint for $n \le N$. Then, if $\eta \in C_c^\infty(B(0,1); [0,1])$ is a smooth cut-off function with compact support, with $\eta \equiv 1$ in a neighborhood of 0, the function

$$\varphi_t^N(x) = \sum_{n=0}^N \eta\left(\frac{x - x_{n,t}}{\rho_n^N}\right) e_{n[2]} \cdot (x - x_{n,t}) \in C_c^\infty(\mathbb{R}^2)$$

(where we denoted $x_{n,t} := x_n + t e_{n[2]}$ for $n \ge 0$) is such that $v_t(x) = \nabla\varphi_t^N(x)$ for $x = x_{n,t}$, $n = 0, \dots, N$. One has, for $x \in \mathbb{R}^2$,

$$\nabla\varphi_t^N(x) = e_{n[2]}\eta\left(\frac{x - x_{n,t}}{\rho_n^N}\right) + e_{n[2]} \cdot \frac{x - x_{n,t}}{\rho_n^N} \nabla\eta\left(\frac{x - x_{n,t}}{\rho_n^N}\right)$$

and since $\|x - x_{n,t}\| \le \rho_n^N$ where $\nabla\eta((x - x_{n,t})/\rho_n^N)$ is not vanishing, $\|\nabla\varphi_t^N(x)\| \le \max\{\|e_1\|, \|e_2\|\}(1 + \|\nabla\eta\|_\infty)$. Hence, one has:

$$\int \|\nabla\varphi_t^N(x) - v_t(x)\|^2 d\mu_t \le \sum_{n > N} a_n \max\{\|e_1\|, \|e_2\|\}^2 (2 + \|\nabla\eta\|_\infty)^2 \to 0$$

as $N \to +\infty$. Notice that possibly reducing the radii $\rho_n^N$, the same construction will produce a function $\varphi_t \in C_c^\infty([0,1] \times \mathbb{R}^d)$ with $\nabla\varphi_t \to v_t$ in $L^2(dt \otimes \mu_t)$ (in both time and space).

*Proof of Proposition 13.* Part (i) follows from Theorem 3 with $\Sigma_0 = \Sigma_1 = \mathrm{Id}$ and $\Sigma_{01} = \Sigma_{10} = -\mathrm{Id}$. Moreover, it holds that

$$\mathcal{L}(v_t | X_0, X_1) = \mathbb{E}[\|v_t((1-t)X_0 + tX_1) - X_1 + X_0\|^2]$$
$$= \mathbb{E}[\|v_t((1-2t)X_0) + 2X_0\|^2] = \mathbb{E}[\| - 2X_0 + 2X_0\|^2] = 0,$$

which shows (ii). For (iii) assume that the ODE $\dot{z}_t(x) = v_t(z_t(x))$ with $z_t(x) = x$ admits a unique solution. Since $v_t$ is locally Lipschitz continuous on $t \in [0, \frac{1}{2})$, this solution is determined by $z_t(x) = (1 - 2t)x$ for $t \in [0, \frac{1}{2})$ which implies that $z_{\frac{1}{2}}(x) = 0$ for all $x$. However, since $z_t$ solves the ODE, we have for all $x \in \mathbb{R}^d$ that

$$-2x = \lim_{\epsilon \to 0+} \frac{z_{\frac{1}{2}}(x) - z_{\frac{1}{2} - \epsilon}(x)}{\epsilon} = \dot{z}_{\frac{1}{2}}(x) = v_{\frac{1}{2}}(z_{\frac{1}{2}}(x)) = v_{\frac{1}{2}}(0) = 0.$$

This is a contradiction. $\qquad\square$

*Proof of Theorem 14.* For the solution of (1) it suffices by (2) to show that $\mathbb{E}[X_1 | X_t = x]$ is smooth in $x$ for all $t < 1$. Using the transformation

$$\begin{pmatrix} X_t \\ X_1 \end{pmatrix} = \begin{pmatrix} (1-t)\mathrm{Id} & \mathrm{Id} \\ 0 & \mathrm{Id} \end{pmatrix} \begin{pmatrix} X_0 \\ X_1 \end{pmatrix},$$

we obtain by the transformation formula the conditional distribution $P_{X_t | X_1 = y}$ has the density

$$p_{X_t | X_1 = y}(x) = \frac{1}{(1-t)^d} p_{X_0 | X_1 = y}\left(\frac{x - ty}{1 - t}\right).$$

Thus, the distribution $\mu_t$ of $X_t$ is absolutely continuous with density

$$p_{X_t}(x) = \int p_{X_t | X_1 = y}\left(\frac{x - ty}{1 - t}\right) d\mu_1(y).$$

Consequently, we obtain by Bayes' theorem that for $x$ within the support of $X_t$ it holds

$$\mathbb{E}[X_1 | X_t = x] = \frac{\int y\, p_{X_t | X_1 = y}(x) d\mu_1(y)}{p_{X_t}(x)} = \frac{\int y\, p_{X_t | X_1 = y}(x) d\mu_1(y)}{\int p_{X_t | X_1 = y}(x) d\mu_1(y)}$$

$$= \frac{\int y\, p_{X_0 | X_1 = y}\left(\frac{x - ty}{1 - t}\right) d\mu_1(y)}{\int p_{X_0 | X_1 = y}\left(\frac{x - ty}{1 - t}\right) d\mu_1(y)},$$

which admits the same smoothness as $p_{X_0|X_1=y}(x)$.

For the solution of (7), note that by the first part $v_t$ is smooth for $t < 1$ (by symmetry, if $P_{X_1|X_0=x_0}$ is smooth with full support for $\mu_0$-a.e. $x_0$, $v_t$ is smooth for $t > 0$). In this case, using the projection property (i) from Proposition 8, we easily derive (1) that for a.e. $t$, $\varphi_t \in H^1_{\text{loc}}(\mathbb{R}^d)$; (2) that $w_t = \nabla \varphi_t$ for such $t$; (3) that $\varphi_t$, as a solution of

$$\Delta \varphi_t = \nabla \ln \mu_t \cdot (v_t - \nabla \varphi_t) + \text{div } v_t$$

is $C^\infty$ in $\mathbb{R}^d$, by a standard bootstrap argument (since for any $k \geq 1$, $\varphi_t \in H^k_{\text{loc}}$ implies $\varphi_t \in H^{k+1}_{\text{loc}}$). In particular, we can integrate $\dot{X} = w_t(X)$ for any $X(0, x) = x$ and build a unique corresponding transport map. This map itself is smooth if both $P_{X_0|X_1=x_1}$ and $P_{X_1|X_0=x_0}$ are smooth and positive. $\qquad\square$

*Proof of Corollary 17.* Note that $(X_0^c, X_1^c) = \mathcal{N}(0, \Sigma)$ for $\Sigma = \begin{pmatrix} \Sigma_0 & \Sigma_{10} \\ \Sigma_{01} & \Sigma_1 \end{pmatrix}$ with $\Sigma_0 = (1 + c^2)\text{Id}$, $\Sigma_1 = \text{Id}$ and $\Sigma_{01} = \Sigma_{10} = -\text{Id}$. Hence, we have by Theorem 3 (i) that the minimizer $v_t^c$ in (1) is given by

$$v_t^c(x) = \frac{(2t-1) - c^2(1-t)^2 - (2t-1)^2}{(1-t)(c^2(1-t)^2 + (2t-1)^2)} x \to -\frac{2}{1-2t} x$$

for $c \to 0$. Since $v_t^c$ is a multiple of the identity, it has a potential such that $v_t^c$ is the minimizer in (7). Moreover we have that for $c \to 0$ it holds

$$\int_0^1 \mathbb{E}[\|v_t^c(X_t^c)\|^2] dt \to \int_0^1 \mathbb{E}\left[\left\|-\frac{2}{1-2t}X_t\right\|^2\right] dt = \int_0^1 \mathbb{E}\left[\|2X_0\|^2\right] dt = 4.$$

Thus for $c$ small enough we have that $\int_0^1 \mathbb{E}[\|v_t^c(X_t^c)\|^2] dt > 4 - \epsilon$. Similarly, we have for $c \to 0$ that

$$\mathbb{E}[\|X_1^c - X_0^c\|^2] = \mathbb{E}[\|X_1 - X_0 - cW\|^2] = \mathbb{E}[\| - 2X_0 - cW\|^2] \to \mathbb{E}[\|2X_0\|^2] = 4$$

such that for $c$ small enough it holds $\mathbb{E}[\|X_1^c - X_0^c\|^2] > 4 - \epsilon$. On the other side it holds that $\mu_0^c = \mathcal{N}(0, (1 + c^2)\text{Id})$ and $\mu_1 = \mathcal{N}(0, \text{Id})$ such that by the explicit form of the Wasserstein distance of two Gaussians we have that $W_2^2(\mu_0^c, \mu_1) \to 0$ for $c \to 0$. Finally, the loss function fulfills

$$\mathcal{L}(v_t^c|X_0^c, X_1^c) = \int_0^1 \mathbb{E}[\|v_t(X_t^c) - X_1^c + X_0^c\|^2] dt \to \int_0^1 \mathbb{E}[\| - \frac{2}{1-2t}X_t - X_1 + X_0\|^2] dt$$

$$= \int_0^1 \mathbb{E}[\| - 2X_0 + 2X_0\|^2] dt = 0.$$

Thus for $c$ small enough the loss is smaller than $\epsilon$. $\qquad\square$

## C  Proof of Theorem 18

*Proof of part (i).* We have that $(X_0^{(i)}, X_1^{(i)})$ is rectifiable by Theorem 14 and Example 16. Moreover, we have by the definition, that $X_0^{(i)}$ is the sum of two independent Gaussian random variables with zero mean and covariances $(1 - c)\text{Id}$ and $c\text{Id}$. Thus, $X_0^{(i)} \sim \mathcal{N}(0, \text{Id})$. Since $\mathcal{R}$ preserves the marginals, this yields the claim. $\qquad\square$

For the proof of the second part, we need the following lemma.

**Lemma 21.** *We have* $\mathbb{E}[\|X_1^{(i)} - X_0^{(i)}\|^2] \leq \mathbb{E}[\|Z_1^{(i)} - Z_0^{(i)}\|^2] + c_i + c_i^2 V_1 + c_i^2$.

*Proof.* Since $W^{(i)}$ is independent of $(Z_0^{(i)}, Z_1^{(i)})$, it holds

$$\mathbb{E}[\|X_1^{(i)} - X_0^{(i)}\|^2] = \mathbb{E}[\|Z_1^{(i)} - \sqrt{1-c_i}Z_0^{(i)} - \sqrt{c_i}W^k\|^2]$$

$$= \mathbb{E}[\|Z_1^{(i)} - \sqrt{1-c_i}Z_0^{(i)}\|^2] + c_i\mathbb{E}[\|W^k\|^2]$$

$$= \mathbb{E}[\|Z_1^{(i)} - \sqrt{1-c_i}Z_0^{(i)}\|^2] + c_i.$$

We bound the remaining term as

$$\mathbb{E}[\|Z_1^{(i)} - \sqrt{1-c_i}Z_0^{(i)}\|^2] = \mathbb{E}[\|Z_1^{(i)} - Z_0^{(i)} - (1-\sqrt{1-c_i})Z_0^{(i)}\|^2]$$
$$= \mathbb{E}[\|Z_1^{(i)} - Z_0^{(i)}\|^2] + (1-\sqrt{1-c_i})^2\mathbb{E}[\|Z_0^{(i)}\|^2]$$
$$- (1-\sqrt{1-c_i})\left(\mathbb{E}[(Z_1^{(i)})^T Z_0^{(i)}]] + \mathbb{E}[\|Z_0^{(i)}\|^2]\right)$$

Using Hölder's inequality and the estimate $1 - \sqrt{1-c_i} \leq c_i$, this is smaller or equal than

$$\mathbb{E}[\|Z_1^{(i)} - Z_0^{(i)}\|^2] + (1-\sqrt{1-c_i})^2\mathbb{E}[\|Z_0^{(i)}\|^2]$$
$$+ (1-\sqrt{1-c_i})\left(\sqrt{\mathbb{E}[\|Z_1^{(i)}\|^2]\mathbb{E}[\|Z_0^{(i)}\|^2]} + \mathbb{E}[\|Z_0^{(i)}\|^2]\right)$$
$$\leq \mathbb{E}[\|Z_1^{(i)} - Z_0^{(i)}\|^2] + c_i\mathbb{E}[\|Z_0^{(i)}\|^2] + c_i^2\left(\sqrt{\mathbb{E}[\|Z_1^{(i)}\|^2]\mathbb{E}[\|Z_0^{(i)}\|^2]} + \mathbb{E}[\|Z_0^{(i)}\|^2]\right)$$

Since by definition it holds that $Z_0^{(i)} \sim \mu_0$ and $Z_1^{(i)} \sim \mu_1$, we have $\mathbb{E}[\|Z_1^{(i)}\|^2] = V_1^2$ and $\mathbb{E}[\|Z_0^{(i)}\|^2] = V_0^2 = 1$, such that the above formula is is equal to

$$\mathbb{E}[\|Z_1^{(i)} - Z_0^{(i)}\|^2] + c_i + c_i^2 V_1 + c_i^2.$$

$\square$

Now, we can proof the second part of Theorem 18.

*Proof of Theorem 18 (ii).* By [34, eqt (28)], we have that

$$\mathbb{E}[\|X_1^{(i)} - X_0^{(i)}\|^2] - \mathbb{E}[\|Z_1^{(i+1)} - Z_0^{(i+1)}\|^2] \geq L^{(i)}.$$

Thus, we get by Lemma 21 that

$$L^{(i)} \leq \mathbb{E}[\|Z_1^{(i)} - Z_0^{(i)}\|^2] - \mathbb{E}[\|Z_1^{(i+1)} - Z_0^{(i+1)}\|^2] + c_i + c_i^2 V_1 + c_i^2$$
$$\leq \mathbb{E}[\|Z_1^{(i)} - Z_0^{(i)}\|^2] - \mathbb{E}[\|Z_1^{(i+1)} - Z_0^{(i+1)}\|^2] + (2 + V_1)c_i.$$

Summing this equation up for $i = 0, ..., K-1$, we obtain

$$\sum_{i=0}^{K-1} L^{(i)} \leq \mathbb{E}[\|Z_1^{(0)} - Z_0^{(0)}\|^2] - \mathbb{E}[\|Z_1^{(K)} - Z_0^{(K)}\|^2] + K(2 + V_1)\bar{c}_K$$
$$\leq \mathbb{E}[\|Z_1^{(0)} - Z_0^{(0)}\|^2] + K(2 + V_1)\bar{c}_K.$$

Noting that the minimum of the $L^{(i)}$ is always smaller or equal than the mean, this yields

$$\min_{i=0,...,K-1} L^{(i)} \leq \frac{\mathbb{E}[\|Z_1^{(0)} - Z_0^{(0)}\|^2]}{K} + (2 + V_1)\bar{c}_K \in \mathcal{O}\left(\frac{1}{K} + \bar{c}_K\right).$$

$\square$

## D  Numerical Verification of the Counterexamples

In this appendix, we numerically verify the findings from Section 4.1. To this end, we consider the iteration $(X_0^{(i+1)}, X_1^{(i+1)}) = \mathcal{R}_p((X_0^{(i)}, X_1^{(i)}))$, which corresponds to minimizing

$$\mathcal{L}(v_t^{(i)}|X_0^{(i)}, X_1^{(i)}) = \int_0^1 \mathbb{E}[\|v_t^{(i)}((1-t)X_0^{(i)} + tX_1^{(i)}) - X_1^{(i)} + X_0^{(i)}\|^2].$$

For the implementation, we parameterize the velocity fields $v_t^{(i)}$ as $v_t^{(i)}(x) = \varphi^{(i)}(t, x)$, where $\varphi^{(i)}$ is a fully connected ReLU neural network with three hidden layers and 512 neurons per hidden layer. We minimize the loss functions $\mathcal{L}(v_t^{(i)}|X_0^{(i)}, X_1^{(i)})$ with the Adam optimizer for 40000 steps with batch size 256 and initial step size $10^{-2}$ which is reduced by a factor of 0.995 every 40 steps. For the initial coupling $(X_0^{(0)}, X_1^{(0)})$, we consider three cases. These are

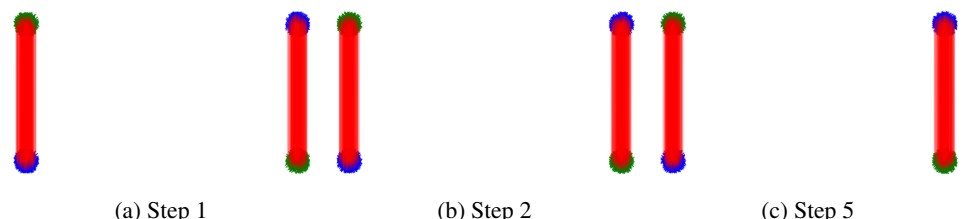

(a) Step 1                     (b) Step 2                     (c) Step 5

Figure 3: Iterative rectification for the example from (11) initialized with the optimal coupling $(X_0, X_1)$.

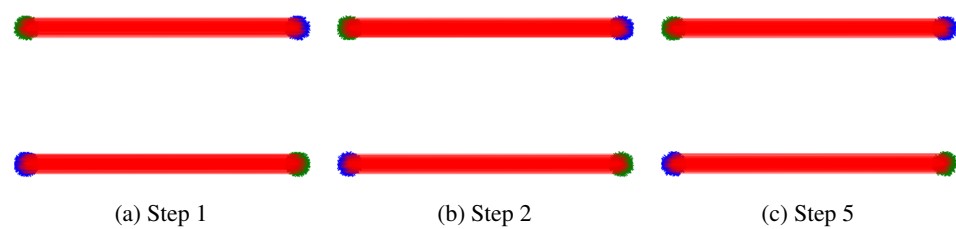

(a) Step 1                     (b) Step 2                     (c) Step 5

Figure 4: Iterative rectification for the example from (11) initialized with the non-optimal fixed point $(\tilde{X}_0, \tilde{X}_1)$ of $\mathcal{R}_p$.

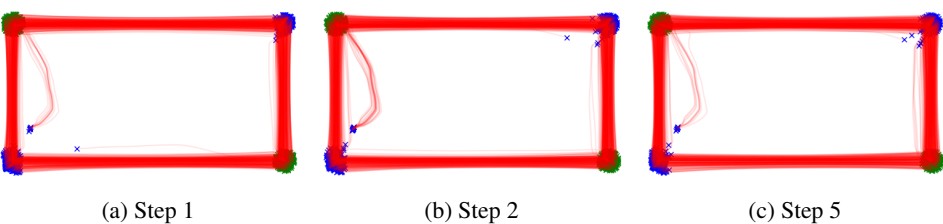

(a) Step 1                     (b) Step 2                     (c) Step 5

Figure 5: Iterative rectification for the example from Section 4.1 initialized with the independent coupling.

- the optimal coupling $(X_0, X_1)$ from (11),

- the non-optimal coupling $(\tilde{X}_0, \tilde{X}_1)$ from (11), and

- the independent coupling, i.e., we choose $X_0^{(0)}$ and $X_0^{(1)}$ to be independent.

The results are given in Figure 3 (optimal coupling), Figure 4 (non-optimal fixed point of $\mathcal{R}_p$) and Figure 5 (independent coupling), where we always plot samples from $X_0^{(i)}$, the trajectory $Z_t^{(i)}$ of $\dot{Z}_t^{(i)} = v_t^{(i)}(Z_t^{(i)})$ and the final samples $X_1^{(i)}$. We plot these results for the first, second and fifth step, i.e., for $i = 1$, $i = 2$ and $i = 5$. The results verify that both couplings from (11) are indeed fixed points of $\mathcal{R}_p$. If we start with the independent coupling, it does not converge to either of them within the first couple of steps. Instead, it splits the mass of both modes from $\mu_0$ and transports it to either of the modes from $\mu_1$. We observe that approximating this non-smooth velocity is hard for neural networks such that numerical errors appear. Nevertheless, it seems to be close to a fixed point of $\mathcal{R}_p$, since the coupling does not change much throughout the iterations.

We redo the experiment for the example from Remark 12. Note that here, we do not have access to the analytical form of the optimal coupling. As a remedy, we generate two datasets of 20000 points from $\mu_0$ and $\mu_1$ and compute the discrete optimal coupling between them using the Python Optimal Transport (POT) package [17]. The results are given in Figure 6 (optimal coupling), Figure 7 (non-optimal fixed point of $\mathcal{R}_p$) and Figure 8 (independent coupling). The black line indicates

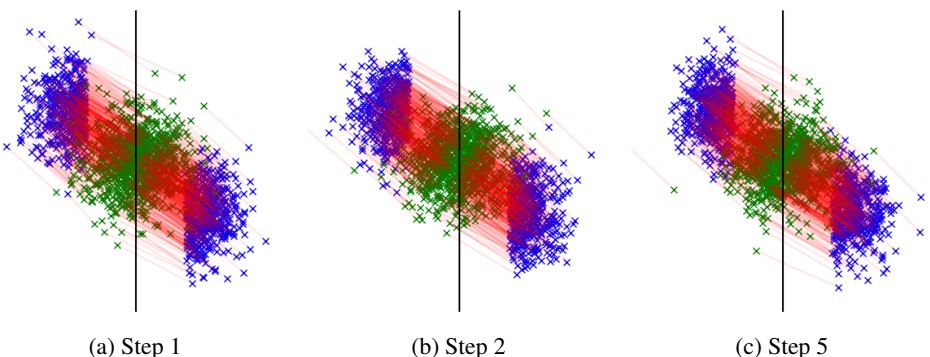

(a) Step 1            (b) Step 2            (c) Step 5

Figure 6: Iterative rectification for the example in Remark 12 initialized with the optimal coupling.

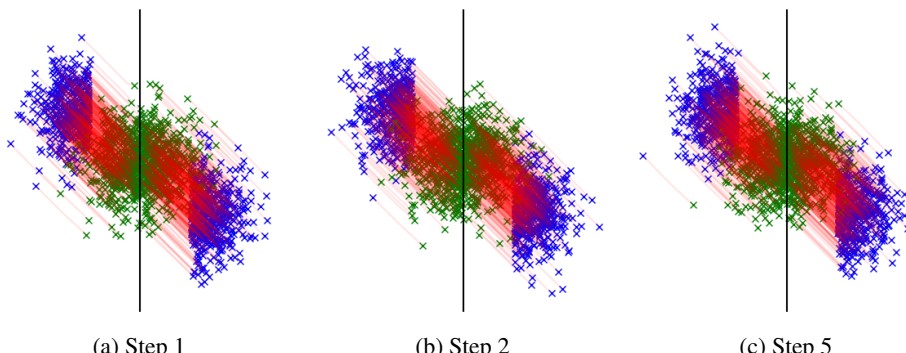

(a) Step 1            (b) Step 2            (c) Step 5

Figure 7: Iterative rectification for the example in Remark 12 initialized with the non-optimal fixed point of $\mathcal{R}_p$ from (12).

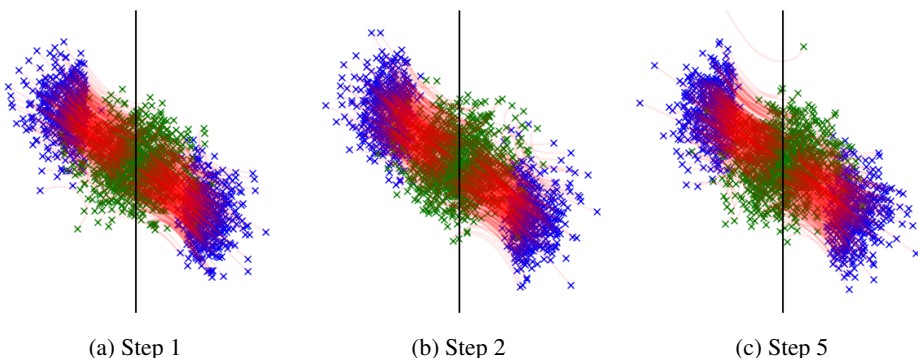

(a) Step 1            (b) Step 2            (c) Step 5

Figure 8: Iterative rectification for the example in Remark 12 initialized with the independent coupling.

Table 1: Caption

| $c_0$ | step 1 | step 2 | step 3 | step 4 | step 5 |
|---|---|---|---|---|---|
| 0 | 2.84 | 2.81 | 2.78 | 2.80 | 2.79 |
| 0.01 | 2.79 | 2.75 | 2.73 | 2.70 | 2.68 |
| 0.05 | 2.73 | 2.68 | 2.69 | 2.68 | 2.67 |
| 0.1 | 2.70 | 2.69 | 2.70 | 2.70 | 2.71 |
| 0.2 | 2.68 | 2.64 | 2.65 | 2.67 | 2.66 |

the separation between the left and right half-plane. We can see that the non-optimal coupling from Remark 12 is indeed a fixed point of $\mathcal{R}_p$ and transports the mass on the left and right half-plane separately. In contrast, the optimal coupling splits the mass differently. If we start with the independent coupling, the process seems to converge to the optimal coupling, even though the lines are not fully straight even after five iterations.

## E    Numerical Experiments for the Smoothed Rectification

We consider the example of Remark 12 and apply the smoothed rectification with parameter $c_k = \frac{c_0}{k}$ for $c_0 \in 0, 0.01, 0.05, 0.1, 0.2$, where $c_0 = 0$ resembles the case of the standard rectification $\mathcal{R}_p$. The initial coupling $(X_0^{(0)}, X_1^{(0)})$ is set to the non-optimal fixed point of $\mathcal{R}_p$ from Remark 12 (defined in eqt (12), visualization in Figure 1c). We report the transport distance $(\mathbb{E}[\|X_0 - X_1\|^2])^{1/2}$ versus the number of steps of the smoothed rectification procedure in Table 1. A lower transport distance indicates that the coupling is closer to optimal transport. The optimal coupling admits a transport distance of 2.66 (evaluated based on 20000 samples using the POT package [17]). Since all experiments are initialized with the same coupling, the first step coincides over all runs, independent of the noise level. Due to numerical errors in the marginals, the transport distance is sometimes slightly smaller than the analytical optimum. Overall, we observe that the smoothed rectification indeed escapes the non-optimal fixed points from Section 4.1. However, the convergence becomes slower if the noise level approaches zero. The generated couplings a visualized in for the different choices of $c_0$ in the Figures 9 to 12.

## F    Relation to Diffusion Schrödinger Bridge Matching

In this appendix, we first briefly describe the relation of rectified flows to Schrödinger bridge matching in Section F.1. While for this method, in contrast to rectified flows, the convergence to the *regularized* optimal transport plan is guaranteed, our counterexamples suggest that the convergence speed becomes arbitrary slow for these examples when the regularization strength tends to zero. We justify this hypothesis numerically in Subsection F.2.

### F.1    Diffusion Schrödinger Bridge Matching

In [1, 46], the authors introduce a noisy version of rectified flows, where the interpolation variables $X_t$ are replaced by noisy versions given as

$$X_t = (1 - t)X_0 + tX_1 + \sqrt{\epsilon t(1 - t)}Z, \quad Z \sim \mathcal{N}(0, I).$$

By denoting the law of $X_t$ by $\mu_t$ and choosing the drift term $v_t(x) = \frac{\mathbb{E}[X_1 - X_t | X_t = x]}{1 - t}$ it can be shown that the Fokker-Planck equation

$$\partial_t \mu_t + \text{div}(v_t \mu_t) = \sqrt{\epsilon}\Delta\mu_t$$

is fulfilled, see [1, 46] for details. In particular, samples from $X_1$ can be generated by sampling from the stochastic differential equation

$$dY_t = v_t(Y_t)dt + \sqrt{\epsilon}dW_t \tag{14}$$

where $W_t$ denotes a Brownian motion. Denoting by $(Z_t)_{t \in [0,1]}$ a solution of this SDE, we obtain the noisy rectification $\mathcal{R}_\epsilon(X_0, X_1) = (Y_0, Y_1)$, where $\epsilon = 0$ recovers the standard rectification $\mathcal{R}$.

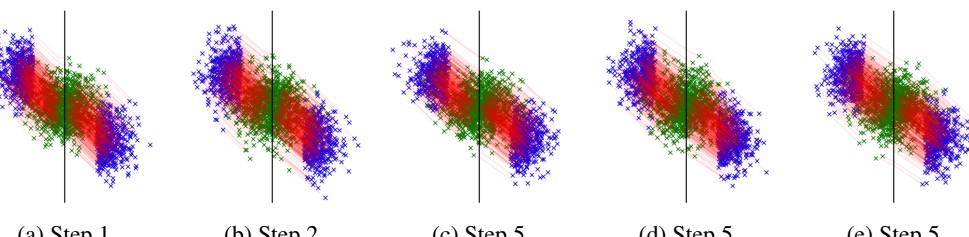

(a) Step 1     (b) Step 2     (c) Step 5     (d) Step 5     (e) Step 5

Figure 9: Smoothed rectification for the example in Remark 12 with $c_0 = 0.01$.

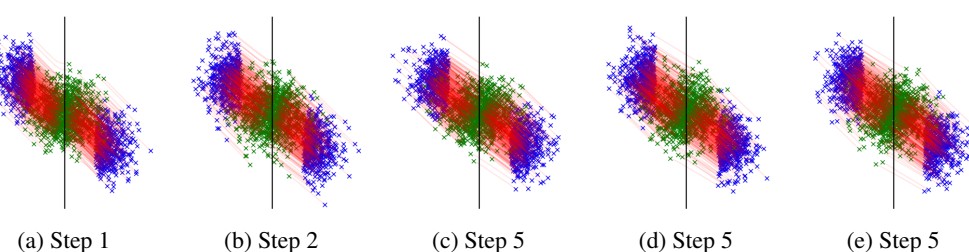

(a) Step 1     (b) Step 2     (c) Step 5     (d) Step 5     (e) Step 5

Figure 10: Smoothed rectification for the example in Remark 12 with $c_0 = 0.05$.

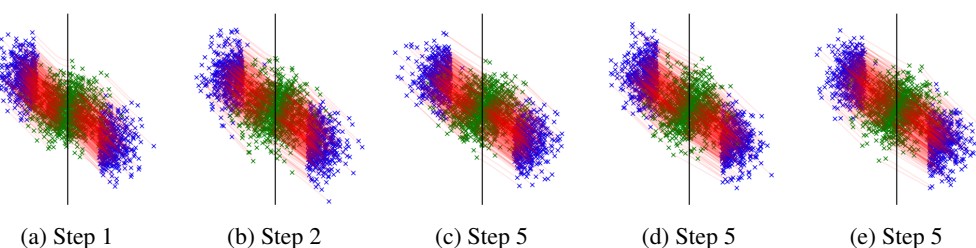

(a) Step 1     (b) Step 2     (c) Step 5     (d) Step 5     (e) Step 5

Figure 11: Smoothed rectification for the example in Remark 12 with $c_0 = 0.1$.

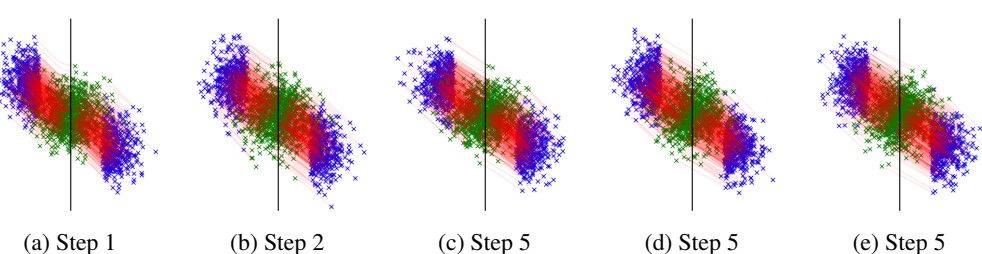

(a) Step 1     (b) Step 2     (c) Step 5     (d) Step 5     (e) Step 5

Figure 12: Smoothed rectification for the example in Remark 12 with $c_0 = 0.2$.

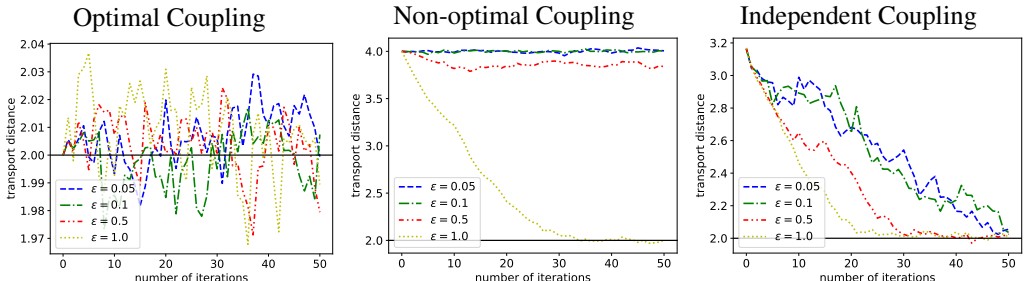

Figure 13: Transport distance for DSBM for the example from Section 4.1 initialized with the coupling $(\tilde{X}_0, \tilde{X}_1)$ from (11) (left), the coupling $(\tilde{X}_0, \tilde{X}_1)$ from (11) (middle) and the independent coupling (right).

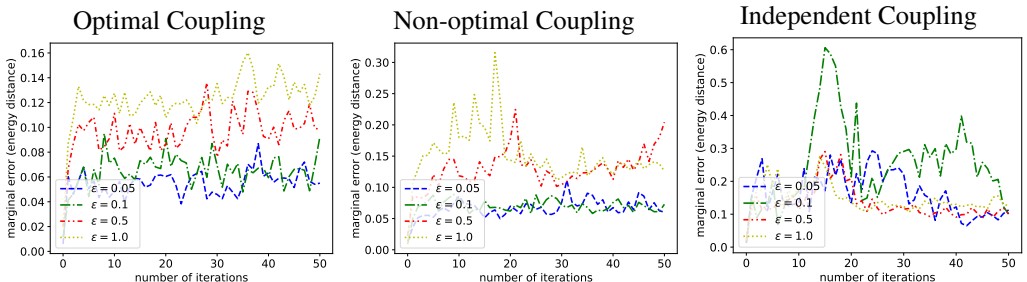

Figure 14: Error introduced in the marginals introduced by errors of DSBM measured in the energy distance for the example from Section 4.1 initialized with the coupling $(\tilde{X}_0, \tilde{X}_1)$ from (11) (left), the coupling $(\tilde{X}_0, \tilde{X}_1)$ from (11) (middle) and the independent coupling (right).

Now, the authors of [46] showed that the iterative noisy rectification given by $\left(X_0^{(i+1)}, X_1^{(i+1)}\right) = \mathcal{R}_\epsilon(X_0^{(i)}, X_1^{(i)})$ converges to a solution of the Schrödinger bridge problem, which is equivalent to entropically regularized optimal transport, see [31] for an overview. Based on this observation, they propose a variation of the rectified flows algorithm, called Diffusion Schrödinger Bride Matching (DSBM), where the velocity field $v_t$ is approximated by a neural network which is then trained by the loss function

$$v_t \in \underset{w_t \in L^2(\mu_t)}{\arg\min} \ \mathcal{L}(w_t|X_0, X_1), \quad \mathcal{L}(w_t|X_0, X_1) \coloneqq \int_0^1 \mathbb{E}\left[\left\| w_t(X_t) - \frac{X_1 - X_t}{1-t} \right\|^2\right] dt. \quad (15)$$

Again, for $\epsilon = 0$ this loss function coincides with the loss function (1) for rectified flows.

In practice, for the sake of numerical stability, the authors of [46] propose to train not only the drift of the SDE (14), but also the time-reversal which is has the drift $w_t(x) = \frac{\mathbb{E}[X_0 - X_t | X_t = x]}{t}$ which leads to an analogous loss function as (15). Note that related generative models for computing Schrödinger bridges were proposed in [15, 41, 50].

### F.2 Numerical Examples with Diffusion Schrödinger Bridge Matching

Next, we numerically investigate the convergence speed of DSBM for our example (11). To this end, we run DSBM for 50 iterations where the initial coupling is given (as in Appendix D) by

- the optimal coupling $(X_0, X_1)$ from (11),
- the non-optimal coupling $(\tilde{X}_0, \tilde{X}_1)$ from (11), and
- the independent coupling, i.e., we choose $X_0^{(0)}$ and $X_0^{(1)}$ to be independent.

We run this test for different regularization parameters $\epsilon \in \{0.05, 0.1, 0.5, 1\}$. For evaluating the results, we plot two error measures versus the number of iterations:

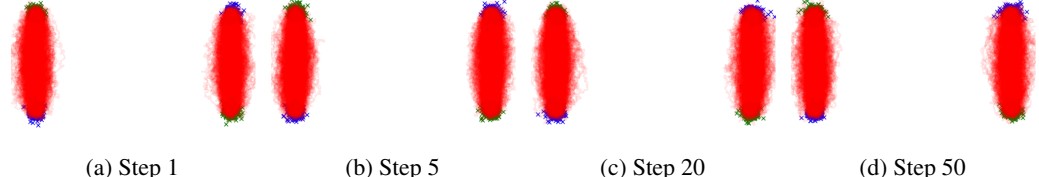

|            |            |            |            |
|------------|------------|------------|------------|
| (a) Step 1 | (b) Step 5 | (c) Step 20 | (d) Step 50 |

Figure 15: Iterative rectification with the DSBM algorithm [46] ($\epsilon = 0.1$) for the example from Section 4.1 initialized with the optimal coupling.

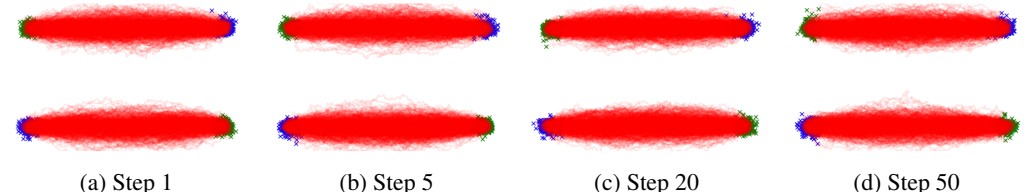

|            |            |            |            |
|------------|------------|------------|------------|
| (a) Step 1 | (b) Step 5 | (c) Step 20 | (d) Step 50 |

Figure 16: Iterative rectification with the DSBM algorithm [46] ($\epsilon = 0.1$) for the example from Section 4.1 initialized with the coupling $(\tilde{X}_0, \tilde{X}_1)$ from (11).

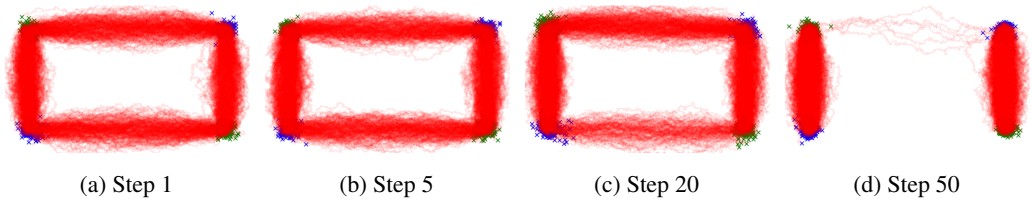

|            |            |            |            |
|------------|------------|------------|------------|
| (a) Step 1 | (b) Step 5 | (c) Step 20 | (d) Step 50 |

Figure 17: Iterative rectification with the DSBM algorithm [46] ($\epsilon = 0.1$) for the example from Section 4.1 initialized with the independent coupling.

- First, we consider the transport cost $\mathbb{E}[\|X_0^{(i)} - X_1^{(i)}\|^2]^{1/2}$ of the coupling $(X_0^{(i)}, X_1^{(i)})$. This quantity measures how close the coupling is to the optimal transport plan.

- Second, we consider the distance of the distributions of $\mu_0^{(i)}$ of $X_0^{(i)}$ and $\mu_1^{(i)}$ of $X_1^{(i)}$ to the original distributions $\mu_0$ and $\mu_1$. To this end, we evaluate the energy distance

$$\mathcal{E}(\mu, \nu) = \left( \int \int \|x - y\| d(\mu - \nu)(x) d(\mu - \nu)(y) \right)^{1/2}$$

between $\mu_0$ and $\mu_0^{(i)}$ and between $\mu_1$ and $\mu_1^{(i)}$. This quantity measures the error which is introduced by the neural network approximation of the drift terms, the optimization error in the loss function and the sampling error in the SDE simulation.

For both evaluation metrics, we discretize the expectation by 50000 samples. The results are given in Figure 13 and 14. Additionally, we plot the corresponding coupling and trajectories for $\epsilon = 0.1$ and iteration $i \in \{1, 5, 20, 50\}$ in Figure 15 (optimal coupling from (11)), Figure 16 (non-optimal coupling from (11)) and Figure 17 (independent coupling). We observe that for our counterexample from Section 4.1 a very large regularization parameter $\epsilon$ is required in order to converge to the entropic optimal transport plan. However, when initialized with the independent coupling DSBM seems to converge in a reasonable time even for moderate $\epsilon$. However, for larger $\epsilon$ also the error introduced in the distributions of $X_0^{(i)}$ and $X_1^{(i)}$ increases. In summary, the examples show that we cannot hope for reasonable *global* convergence rates of the DSBM algorithm. Whether such rates can be derived *locally*, remains an open question.

