# OpenReview forum: "On the Relation between Rectified Flows and Optimal Transport"
_NeurIPS.cc/2025/Conference — NeurIPS 2025 poster_

### Official Review · Reviewer_u4yX · 2025-06-30

**Clarity:** 2
**Significance:** 2
**Originality:** 3
**Rating:** 3
**Confidence:** 4

**Summary:**

This paper re-examines the relationship between rectified flows and optimal transport (OT), arguing decisively that rectified flows, even with gradient constraints, are not a reliable method for computing OT solutions. The authors substantiate this claim through counterexamples, including a key scenario where distributions on disconnected supports cause a non-optimal transport plan to become a fixed point of the iterative rectification process. They further show that a "non-rectifiable" coupling can achieve zero loss without being optimal, demonstrating that a vanishing training objective does not guarantee optimality. Beyond these refutations, the paper provides a foundational analysis of rectified flows—deriving invariance properties, giving explicit solutions for Gaussian cases, and establishing conditions for rectifiability. Finally, to address the identified issues with ill-posed dynamics, the authors propose a "smoothed rectification" procedure that injects noise at each iteration to ensure the transport problem remains well-defined.

**Questions:**

Does noise injection also help the iterative process escape these non-optimal fixed points? For instance, in your disconnected support counterexample (Fig. 1), if one were to run the smoothed rectification procedure starting from the independent coupling, would it be more likely to converge to the globally optimal transport plan (Fig. 1a) compared to the standard procedure? My evaluation of the paper's practical contribution would be further strengthened if the authors could provide either (a) an empirical study, perhaps by extending the experiments in Appendix B, to show that smoothing helps find the correct OT plan, or (b) a theoretical argument or conjecture on how noise might provide a regularizing effect that favors the true optimum.

**Ethical Concerns:**

["NO or VERY MINOR ethics concerns only"]

**Final Justification:**

This paper studies some basic properties of reflow and reveals its relations to optimal transport. While theoretically sound, it's hard to see the real-world value without using standard benchmarks. Instead of just using examples like Gaussian noise, the authors should use common datasets like MNIST and ImageNet to justify their theory.

So I stick to my original score.

**Limitations:**

Yes

**Paper Formatting Concerns:**

No formatting concerns

**Quality:**

3

**Strengths And Weaknesses:**

Strengths
1. The core contribution—the systematic refutation of the equivalence between rectified flows and OT—is original. The idea of linking failure modes to disconnected supports and non-rectifiable couplings is a novel perspective.

2. The paper has good technical quality. The arguments are backed by mathematical proofs and precisely constructed counterexamples.

3. The paper has good contribution. By demonstrating the limitations of using rectified flows for optimal transport, it may clarify misconceptions and prevents the community from pursuing a flawed direction.

Weaknesses
1. While the proposed "smoothed rectification" method is a valuable idea, its analysis feels preliminary. The theoretical guarantees are shown for the specific case where the source distribution is Gaussian, which may not cover all use cases. More importantly, while the authors prove that their method maintains a good convergence rate for the *loss*, they do not investigate if it actually helps in converging to the *true optimal transport plan*, especially in the challenging cases (like disconnected supports) that they identify.

2. While the counterexample in Section 4.1 based on disconnected supports is technically sound and arguably the paper's most critical contribution, its practical significance is not immediately obvious from the text. The authors should explicitly state why this scenario is not just a theoretical curiosity but a common and fundamental challenge in practice.

3.No experiments demonstrate the proposed smoothed rectification procedure.

---

> ### Author Rebuttal · Authors · 2025-07-29
>
> Many thanks for your reading of our paper, and your review, comments and suggestions! We reply to all of your critiques separately.
>
> > While the proposed "smoothed rectification" method is a valuable idea, its analysis feels preliminary. The theoretical guarantees are shown for the specific case where the source distribution is Gaussian, which may not cover all use cases. More importantly, while the authors prove that their method maintains a good convergence rate for the loss, they do not investigate if it actually helps in converging to the true optimal transport plan, especially in the challenging cases (like disconnected supports) that they identify.
>
> Please find our answer to this comment together with the answer to your last comment at the end of our reply.
>
> > While the counterexample in Section 4.1 based on disconnected supports is technically sound and arguably the paper's most critical contribution, its practical significance is not immediately obvious from the text. The authors should explicitly state why this scenario is not just a theoretical curiosity but a common and fundamental challenge in practice.
>
> One key motivation and application of rectified flows is distribution translation, i.e., constructing a mapping that transports one probability distribution to another. In this setting, connecting rectified flows to **optimal transport** is particularly appealing, as it yields a mapping that is **unique** (under absolute continuity), **stable**, and **interpretable** (minimizing the expected squared distance between inputs and outputs).
>
> However, our counterexample demonstrates that this connection **breaks down when the distributions are disconnected**. Since disconnected datasets are common in practice, this indicates that **rectified flows cannot reliably compute optimal transport mappings** in general.
>
> Another major motivation for rectified flows is the fact that the flow paths tend to become "straighter", improving computational efficiency during sampling by simplifying the ODE trajectories. This aspect is largely independent of optimal transport, and our counterexample does not affect this advantage.
>
> In the next revision, we will extend the comments in introduction, Section 4.1 and Conclusions based on this discussion.
>
> > No experiments demonstrate the proposed smoothed rectification procedure.
>
> In the next revision, we will extend Appendix B by some numerical experiments in the setting of Remark 9 (see our reply to the next comment, a brief overview over these experiments is given at the end of the rebuttal).
>
> > Does noise injection also help the iterative process escape these non-optimal fixed points? For instance, in your disconnected support counterexample (Fig. 1), if one were to run the smoothed rectification procedure starting from the independent coupling, would it be more likely to converge to the globally optimal transport plan (Fig. 1a) compared to the standard procedure? My evaluation of the paper's practical contribution would be further strengthened if the authors could provide either (a) an empirical study, perhaps by extending the experiments in Appendix B, to show that smoothing helps find the correct OT plan, or (b) a theoretical argument or conjecture on how noise might provide a regularizing effect that favors the true optimum.
>
> From a theoretical side, the definition of the smoothed rectification ($X_0^{(i)}=\sqrt{1-c_i}Z_0^{(i)}+\sqrt{c_i}W^{(i)}$ for standard Gaussian noise $W^{(i)}$) ensures not only rectifiability, but also that $\mathrm{supp}(X_t^{(i)})=\mathbb R^{d}$ (for $t<1$). Consequently, the smoothed rectification indeed overcomes the problem with disconnected supports which is the main idea for the counterexamples from Section 4.1. However, while the smoothed rectification helps for the counterexamples from Section 4.1, we are currently unable to provide a proof that it generally converges to the optimal coupling (for appropriate $c_i$).
>
> We have now also tested the smoothed rectification for the counterexample from Remark 9 (we do not use the other example from Section 4.1, because the smoothed rectification assumes that $\mu_0$ is Gaussian). As initialization, we use the non-optimal fixed point from eqt (12). The results (see below) suggest that smoothed rectification does converge to the optimal coupling. Furthermore, the convergence speed significantly decreases as the noise level approaches zero. This is expected since a small change in the coupling results only in the small change in the rectification.
>
> We will present these findings and the corresponding discussion in Appendix B of the next revision. A brief overview of the experiments and results is given below (for the paper we will also generate figures in the style of Figure 6/7/8 visualizing the resulting couplings).
>
> ---
>
> ### Additional Experiment with Smoothed Rectification
>
> We consider the example of Remark 9 and apply the smoothed rectification with parameter $c_k=\frac{c_0}{k}$ for $c_0\in\{0,0.01,0.05,0.1,0.2\}$, where $c_0=0$ resembles the case of the standard rectification $\mathcal R_p$. The initial coupling $(X_0^{(0)},X_1^{(0)})$ is set to the non-optimal fixed point of $\mathcal R_p$ from Remark 9 (defined in eqt (12), visualization in Figure 1c). We report the transport distance $(\mathbb E[\\|X_0-X_1\\|^2])^{1/2}$ versus the number of steps of the smoothed rectification procedure in the table below. A lower transport distance indicates that the coupling is closer to otpimal transport. The optimal coupling admits a transport distance of $2.66$ (evaluated based on $20000$ samples using the POT package). Since all experiments are initialized with the same coupling, the first step coincides over all runs, independent of the noise level. Due to numerical errors in the marginals, the transport distance is sometimes slightly smaller than the anlytical optimum. Overall, we observe that the smoothed rectification indeed escapes the non-optimal fixed points from Section 4.1. However, the convergence becomes slower if the noise level approaches zero.
>
> In the paper, we will also include figures (similar to Fig 6/7/8) for this experiments.
>
> $c_0$ | step 1 | step 2 | step 3 | step 4 | step 5
> ------|--------|--------|--------|--------|-------
> $0$   | $2.84$ | $2.81$ | $2.78$ | $2.80$ | $2.79$
> $0.01$| $2.79$ | $2.75$ | $2.73$ | $2.70$ | $2.68$
> $0.05$| $2.73$ | $2.68$ | $2.69$ | $2.68$ | $2.67$
> $0.1$ | $2.70$ | $2.69$ | $2.70$ | $2.70$ | $2.71$
> $0.2$ | $2.68$ | $2.64$ | $2.65$ | $2.67$ | $2.66$

---

> > ### Comment · Reviewer_u4yX · 2025-08-03
> >
> > Thank you for your comprehensive rebuttal.
> >
> > Regarding the smoothed rectification experiments:
> > The experimental table showing convergence to near-optimal transport distances across different noise levels provides empirical support for your method.
> >
> > Regarding the practical significance of disconnected supports:
> > While I understand your motivation for studying distribution translation, I remain unconvinced that
> > truly disconnected datasets are common in practice. The distinction between the underlying
> > distributions and sampled data points is important here. A more compelling demonstration would be
> > a real-world task where smoothed rectification outperforms standard rectification.
> >
> > I maintain my original assessment that the paper makes valuable theoretical contributions, but the
> > practical solution needs more convincing validation.

---

> > > ### Author Response · Authors · 2025-08-05
> > >
> > > Thank you for your reply and additional comment.
> > >
> > > We emphasize that our paper focuses on the theoretical analysis of rectified flows. The major claim of our paper is that we **cannot guarantee convergence of rectified flows to optimal transport** which is clearly demonstrated by our counterexamples. Developing a state-of-the-art generative model (or data-translation model) is beyond the scope of our work.
> > >
> > > Additionally, we respectfully disagree with the reviewers' assertion that commonly used datasets are generally connected. For instance, for MNIST, we do not think that there is a "continuous path" connecting the digits "1" and "5", and we believe that this dataset is disconnected.  In the same way, it is unlikely that there is a path between the faces with glasses and the faces without glasses in the CelebA dataset or the brain images and knee images in FastMRI.
> > >
> > > While we acknowledge that these are benchmark datasets and not necessarily representative for real-world data, we see no reason to assume that application-specific datasets are inherently connected.

---

### Official Review · Reviewer_pPGx · 2025-07-02

**Clarity:** 3
**Significance:** 2
**Originality:** 2
**Rating:** 5
**Confidence:** 3

**Summary:**

The authors of this paper focus on further study of rectified flows, and in particular their relation to optimal transport. Rectified flows straighten the paths of a coupling (transport plan in the language of optimal transport) between two distributions. Given that the paths generated by the velocity field of an optimal transport (in the euclidean case) are straight, the goal is to learn an optimal transport through this rectification. The authors show by providing counterexamples that the assumptions for this to hold are stronger than in a previous paper. They also prove invariances for rectified flows under affine transformations, find formulas for rectified flows (which are optimal transports) in the gaussian case, and propose to smooth the starting distribution at each rectification step to ensure that the couplings stay rectifiable.

**Questions:**

1. Isn't there a mistake in the definition (12) of $X_1$ in Remark 9? Shouldn't the second case also contain $X_0$ instead of $X_1$?
2. To make sure that I understand the smoothed rectification in Theorem 15, you assume $\mu_0$ is gaussian, and since rectification preserves the marginals, $X_0$ will remain gaussian, but it is the conditional distribution of $X_0$ (given $X_1$) that must be absolutely continuous with a smooth and positive density, which is why you add an independent Gaussian noise?

**Ethical Concerns:**

["NO or VERY MINOR ethics concerns only"]

**Final Justification:**

This paper proves useful results about the relation between rectified flows and optimal transport. In my opinion this outweighs the lack of experimentation on large datasets. Furthermore, the rebutal and proposed changes do indeed improve the paper. Therefore I recommend to accept this paper.

**Limitations:**

1. Counterexamples are provided which mean that stronger assumptions are needed for rectification to converge to an optimal transport, but these stronger assumptions are not investigated. This is of course minor as I think the paper is rich enough in results.

**Paper Formatting Concerns:**

No formatting concerns

**Quality:**

3

**Strengths And Weaknesses:**

Strengths:
1. Many theoretical contributions to the study of rectified flows

Weaknesses:
1. To improve clarity, Definitions should be included. For example, the rectified coupling is defined on line 110 with a reference to equation (1). I think the paper would have been easier to follow if such important definitions were in Definition sections such as theorems and propositions.

---

> ### Author Rebuttal · Authors · 2025-07-29
>
> We would like to thank the reviewer for having read carefuly our paper and their efforts in evaluating it. We provide answers to the specific critiques separately.
>
> > To improve clarity, Definitions should be included. For example, the rectified coupling is defined on line 110 with a reference to equation (1). I think the paper would have been easier to follow if such important definitions were in Definition sections such as theorems and propositions.
>
> Thank you for the suggestion! We will revise the paper and put important definitions (like the rectification) into a definition environment.
>
>  > Isn't there a mistake in the definition (12) of $X_1$ in Remark 9? Shouldn't the second case also contain instead $X_0$ of $X_1$?
>
> You are absolutely right! Thank you for pointing out this typo.
>
> > To make sure that I understand the smoothed rectification in Theorem 15, you assume $\mu_0$ is gaussian, and since rectification preserves the marginals, $X_0$ will remain gaussian, but it is the conditional distribution of $X_0$ (given $X_1$) that must be absolutely continuous with a smooth and positive density, which is why you add an independent Gaussian noise?
>
> Correct! Adding independent Gaussian noise (and rescaling $Z_0^{(i)}$) in the definition to $X_0^{(i)}$ ensures that the conditional distribution $P_{X_0|X_1=x_1}$ has a smooth positive density such that we can apply Theorem 11. At the same time $X_0$ remains a Gaussian distribution since the rectification preserves the marginals and a sum of two Gaussians remains Gaussian.
>
> > Counterexamples are provided which mean that stronger assumptions are needed for rectification to converge to an optimal transport, but these stronger assumptions are not investigated. This is of course minor as I think the paper is rich enough in results.
>
> Well, we can at least state sufficient assumptions such that Theorem 5.6 from [1] is correct (where the additional assumption is marked italic):
>
> Assume that $(X_0,X_1)$ is rectifiable and let $v_t=\nabla \varphi_t\in\mathrm{argmin}_{w_t=\nabla \psi_t}\mathcal L(w_t|X_0,X_1)$ fulfill that $\varphi_t\in C^{2,1}(\mathbb R^d\times[0,1])$. _Moreover, suppose that $\mathrm{supp}(X_t)=\mathbb R^d$ for $X_t=(1-t)X_0+tX_1$_. Then the following are equivalent:
> - $\mathcal R_p(X_0,X_1)=(X_0,X_1)$,
> - $\mathcal L(v_t|X_0,X_1)=0$,
> - $(X_0,X_1)$ is an optimal coupling.
>
> However, we stress that within the iterative rectification $(X_0^{(i+1)},X_1^{(i+1)})=\mathcal R(X_0^{(i)},X_1^{(i)})$ it is usually intractable to check, whether these assumptions remain fulfilled for all $i$ based on the initial coupling $(X_0^{(0)},X_1^{(0)})$. Moreover, we showed in Corollary 14 that the fact that the loss decays does not necessary imply convergence of the (couplings) to a fixed point of $\mathcal R$. How (or if at all) the decay of the loss relates to convergence of the couplings to a fixed point remains an open question. In the next revision, we will include the statement of Theorem 5.6 from [1] with the corrected assumptions (also considering the comments of reviewer 3EWA) and extend the explanations on this point in the conclusions.

---

> > ### Comment · Area_Chair_5QCo · 2025-08-05
> >
> > Dear reviewer,
> >
> > Please read the authors' rebuttal if you haven't done so and state your response accordingly.
> >
> > Best,
> > AC

---

> > ### Comment · Reviewer_pPGx · 2025-08-05
> > **Official comment**
> >
> > I appreciate the author's response and indeed agree that the changes they propose to make to the revision will improve the paper and make it clearer

---

> > > ### Author Response · Authors · 2025-08-05
> > > **Thank you**
> > >
> > > Dear reviewer, thank you for your answer.

---

### Official Review · Reviewer_3EWA · 2025-07-02

**Clarity:** 2
**Significance:** 2
**Originality:** 2
**Rating:** 5
**Confidence:** 3

**Summary:**

The authors clarify the relationship between couplings obtained from rectified flows and optimal transport in the setting of vector fields induced by potentials. In particular, they demonstrate via counter-examples, that fixed points and zero-minimisers of the rectified flow objective do not necessarily obtain optimal couplings. This corrects a previous result in the literature which claims the opposite. Moreover, the authors clarify conditions when solutions to the rectified flow objective are indeed rectifiable, and propose a procedure for ensuring rectifiability.

**Questions:**

To what extend can the counter-examples/proofs/intuition from the paper be generalised to rectified flows under the setting of a generic cost function, as presented in [1]?

[1] Qiang Liu (2022) Rectified Flow: A Marginal Preserving Approach to Optimal Transport

**Ethical Concerns:**

["NO or VERY MINOR ethics concerns only"]

**Final Justification:**

The authors have been able to address my concerns from the review making me increase my score to 5 for acceptance. I believe this will be a useful addition to the literature to help clarify misconception about rectified flows.

**Limitations:**

Yes

**Quality:**

3

**Strengths And Weaknesses:**

Strengths:

- While the main contribution presented in the paper is probably already known to theorists in this area, due to the simplicity of the counter-examples presented in the paper, the paper is still a nice addition to the literature for providing a single citable reference, for newcomers to the area, in clarifying the extent to which rectified flows can recover optimal transport (as well as correcting a previous mistake in the literature).

Weaknesses:

- The structure of the paper is a bit confusing. For instance, what is the purpose of Section 3.1? This section appears abruptly from the previous discussion of rectified flows with the gradient potential conditions, and appears to have little connection with the rest of the presentation. From inspection, it appears that Section 3.1 is only used in the proofs of later results. The layout disrupts from the presentation of the main argument of the paper and makes the paper harder to follow.
- Moreover, the paper is a bit overstuffed with off-hand remarks of results which have no accompanying justification. For example:
    - Line 193: "$v^{A,b}_t$ has a potential if and only if the Jacobian is symmetric." - what is the justification of this? I'm not sure how this immediately follows?
    - Line 244: "The statement of...". The full statement of the corrected proposition and proof would be valuable to include in this paper to resolve any ambiguity, instead of leaving this offhand remark halfway in the paper
- For Theorem 2 and Theorem 4, covariance matrices are referred to as both $\Sigma_0$ and $\Sigma_{00}$. Why is this inconsistent?
- In line 184, I do not understand how the independent coupling of $(X_0, X_1)$ follows a GMM distribution as $X_1$ follows a discrete measure. Hence, I do not see how Theorem 4 can be applied to this case.
- The notation in Remark 5 is confusing. I assume that here $\varphi_t^a$ is referring to the transformation by adding the constant $a$. However, in Theorem 1, the notation for this transformation uses the letter $b$ in $v_t^b$. I don't see any reason for changing up the notation here, apart from to reduce clarity.

---

> ### Author Rebuttal · Authors · 2025-07-29
>
> Thank you for having read accurately our paper and for your comments! We reply to all of these separately.
>
> > While the main contribution presented in the paper is probably already known to theorists in this area [...]
>
> We are not aware of any comparable results from the literature. In addition, the slightly incorrect results which we discuss are cited in plenty of papers, without any discussion of their validity. We really feel that pointing this out is very useful to the community.
>
> > The structure of the paper is a bit confusing. For instance, what is the purpose of Section 3.1? This section appears abruptly from the previous discussion of rectified flows with the gradient potential conditions, and appears to have little connection with the rest of the presentation. From inspection, it appears that Section 3.1 is only used in the proofs of later results. The layout disrupts from the presentation of the main argument of the paper and makes the paper harder to follow.
>
> Some relations between rectified flows and optimal transport are already true for the case without the gradient potential condition. Section 3.1 gathers these foundational results, including
>
> - Rectified flows and optimal transport share some invariance properties (Theorem 1 (ii) and (iii))
> - The observation that, in both the Gaussian setting (Theorem 2 (ii)) and the one-dimensional case (Proposition 3), the optimal coupling is achieved after a single rectification step.
>
> As a side effect, these results are quite useful to understand the basic behavior of rectified flows and are consequently used in the proofs of subsequent results.
>
> The current structure aims to provide a general overview of the literature in Section 2, followed by our contributions in the subsequent sections. However, we recognize that a revised layout could enhance accessibility from a pedagogical perspective. In the next revision, we consider to restructure Sections 2 and 3 as follows:
>
> - Section 2 will present the general background, excluding the discussion on the gradient potential condition (currently lines 88–119), and incorporate the results presently found in Section 3.1.
> - Section 3 will begin with the background on the gradient potential condition (lines 120–137), followed by the results currently contained in Section 3.2.
>
> > Line 193: "$v^{A,b}_t$ has a potential if and only if the Jacobian is symmetric." - what is the justification of this? I'm not sure how this immediately follows?
>
> Here we refer to the fact that a vector field admits a potential (locally or in a "simple" topology) if and only if the Jacobian is symmetric (such vector fields are often referred as "conservative"). This is a non-trivial, yet well-known result, contained in many text books: One direction immediately follows from the fact that the Jacobian of the vector field is the Hessian of the potential and consequently symmetric (known as Schwarz's theorem). The other direction is known as Poincaré's lemma (and equivalent to the property that the circulation of such field along a closed loop vanishes). In the next revision, we will add an additional sentence that we use the above equivalence and add a reference to some text book.
>
> > Line 244: "The statement of...". The full statement of the corrected proposition and proof would be valuable to include in this paper to resolve any ambiguity, instead of leaving this offhand remark halfway in the paper
>
> We agree that presenting the corrected statement could be beneficial. In the next revision, we will include the following version of Thm 5.6 from [1] for $c(x-y)=\\|x-y\\|^2$ (where the additonal assumption is marked italic):
>
> Assume that $(X_0,X_1)$ is rectifiable and let $v_t=\nabla \varphi_t\in\mathrm{argmin}_{w_t=\nabla \psi_t}\mathcal L(w_t|X_0,X_1)$ fulfill that $\varphi_t\in C^{2,1}(\mathbb R^d\times[0,1])$. _Moreover, suppose that $\mathrm{supp}(X_t)=\mathbb R^d$ for $X_t=(1-t)X_0+tX_1$_. Then the following are equivalent:
> - $\mathcal R_p(X_0,X_1)=(X_0,X_1)$,
> - $\mathcal L(v_t|X_0,X_1)=0$,
> - $(X_0,X_1)$ is an optimal coupling.
>
> The proof is mainly the same as in [1]. But we will add an appendix, which provides a more detailed explanation on this point.
>
> > For Theorem 2 and Theorem 4, covariance matrices are referred to as both $X_0$ and $X_{00}$. Why is this inconsistent?
>
> This is a typo, thank you for pointing this out. We will correct it within the next revision.
>
> > In line 184, I do not understand how the independent coupling of $(X_0,X_1)$ follows a GMM distribution as $X_1$ follows a discrete measure. Hence, I do not see how Theorem 4 can be applied to this case.
>
> Here we use the definition of the multivariate normal distribution which includes singular covariance matrices. More precisely, we say that a $d$-dimensional random variable $X$ follows normal distribution if and only if all one-dimensional projections $a^T X$ follow a normal distribution with the notation that $\delta_x=\mathcal N(x,0)$ for $x\in\mathbb{R}$. It can be shown that using this definition, Gaussian distributions can still be identified by their mean and covariance matrix (which is still positivie semi-definite but not necessarily invertible) and that it preserves all computation rules of Gaussian distributions which are used within the statement and proofs of the Theorem 2 and 4. Then, the joint distribution of independent $X_0\sim\mathcal N(0,\mathrm{Id})$ and $X_1\sim\sum_{k=1}^K\frac1K\delta_{m^k}$ follows the GMM
> $$\sum_{k=1}^K\frac1K\mathcal N\left(\left(\begin{array}{c}0\\\\m^k\end{array}\right),\left(\begin{array}{cc}\mathrm{Id}&0\\\\0&0\end{array}\right)\right)$$
> In the next revision, we will add a remark about this and add a reference to some text book introducing Gaussian distributions and their computation rule for singular covariance matrices.
>
> > The notation in Remark 5 is confusing. I assume that here $\varphi^a_t$ is referring to the transformation by adding the constant $a$. However, in Theorem 1, the notation for this transformation uses the letter $b$ in $v_t^b$. I don't see any reason for changing up the notation here, apart from to reduce clarity.
>
> This is again a typo, which we will correct in the next revision. We are sorry for the confusion!
>
> > To what extend can the counter-examples/proofs/intuition from the paper be generalised to rectified flows under the setting of a generic cost function, as presented in [1]?
>
> The counterexample regarding the non-rectifiable transport plan from Section 4.2 works for any cost function fulfilling the assumptions from [1]. The main argument here is that the distribution $\mu_t$ of $X_t=(1-t)X_0+tX_1$ becomes singular for $t=\frac12$. Therefore, for $v_t$ fulfilling the continuity equation $\partial_t \mu_t+\mathrm{div}(v_t\mu_t)=0$ (which is true for the minimizer of the loss function) and for any solution of $\dot Z_t=v_t(Z_t)$ with $Z_0=X_0$, it holds that $Z_{1/2}=0$ almost surely. In other words, almost-every ODE path collapses to a single point at time $\frac12$ which contradicts the uniqueness of solutions.
>
> The counterexample from Section 4.1 might not work "as-is" for other cost funcitions (already because it might be possible to construct cost functions such that both couplings from eqt (11) admit the same cost). However, it should be straight-forward to construct analogous examples for other cost functions (which are convex and have a $C^1$ convex conjugate) by the same idea: Construct two locally optimal transport plans which are sepatated by some area where the coupling has zero-mass. Then, glue them together such that the global plan is not optimal.
>
> Within the next revision, we will add a remark about other cost functions in the Conclusions section.
>
>
> Reference:
> [1] Qiang Liu (2022) Rectified Flow: A Marginal Preserving Approach to Optimal Transport

---

> > ### Comment · Area_Chair_5QCo · 2025-08-05
> >
> > Dear reviewer,
> >
> > Please read the authors' rebuttal if you haven't done so and state your response accordingly.
> >
> > Best,
> > AC

---

> > ### Comment · Reviewer_3EWA · 2025-08-05
> >
> > Thank you for the detailed response. This has addressed my concerns raised in the review. I will raise my rating to 5 to recommend acceptance.

---

> > > ### Author Response · Authors · 2025-08-06
> > > **Thank you**
> > >
> > > Thank you very much for your answer.

---

### Decision · Program_Chairs · 2025-09-17

**Decision:**

Accept (poster)

**Comment:**

During the rebuttal the authors have addressed several concerns raised by the reviewers and all the reviewers agree that the paper has merits especially on the theoretical side. I have also assigned an additional reviewer (see the comments below) who also provided a positive evaluation. I recommend an acceptance.

## Additional review
-----
Summary

The broad context of this work is flow models, i.e., for a given source $\mu_0$ and target $\mu_1$ probability measures, the process of learning a time-continuous vector field $v_t: \mathbb R^d \rightarrow  \mathbb  R^d$ which satisfies an ordinary differential equation (ODE) and transforms a sample $X_0 \sim \mu_0$ such that the target $X_1 = v_1 (X_0)$ at the final time $t=1$ is distributed according to $\mu_1$. The pushforward of $\mu_0$ by $v_t$ generates a time-dependent probability density $\mu_t$ (probability path).  Initially, flow models were trained by solving the ODE (simulation) and maximizing the likelihood of the target training examples. Subsequently, training by so-called flow matching was introduced - it relies on parametric models of $v_t$ generating the probability path $\mu_t$ learned by regression This approach avoids simulation and makes training more computationally tractable, and lead to state-of-the art diffusion and other generative models.

More specifically, this work focuses on so-called rectified flows defined as follows. Given a joint probability distribution, called coupling, $\pi$ of $X_0$ and $X_1$ with marginals $X_0 \sim \mu_0$ and $X_1 \sim \mu_1$, we define the interpolation $X_t = (1-t) X_0 + X_1$ and let $\mu_t$ be the law of $X_t$.  A rectified flow is a minimizer of

$$(1) ~\mathcal L (v_t|X_0, X_1) = \int_0^1 \mathbb E \| v_t (X_t)-X_1 +X_0\|^2 dt.$$

where $v_t$ belongs to the space of $L^2(\mu_t)$ functions.  Previous work showed that such minimizer exists, is unique and satisfies the continuity equation
$$(2) ~\partial_t \mu_t + div (v_t \mu_t) = 0 $$
in the distributional sense. Based on that $v_t$ is a feasible vector-valued measure in the minimization problem defining the hydrodynamic formulation of the classic Wasserstein-2 ($W_2$) distance (Brenier-Benamou). If $v_t$ is smooth enough, $v_t$ defines a transport map from $\mu_0$ to $\mu_1$ given by $z_1$ where $z_t$ solves the ODE $\dot z_t(x) = v_t(z_t(x))$ with $z_0 = x$. If this ODE has a unique solution, then the underlying coupling $\pi$  is called "rectifiable" and one can sample $\mu_1$ by sampling $\mu_0$ and solving this ODE. Moreover, it is possible to construct a series of coupling $( X^k_0, X_1^k) $ such that $\min_{k \in [K]} \mathcal L (v_t|X^k_0, X^k_1) \rightarrow 0$ as $K$ get large. The time-dependent velocity $v_t$ corresponding to the optimal coupling being the "rectified" flow.

Previous work showed that even if $\mathcal L (v_t|X_0, X_1) =0$, the corresponding $v_t$ is not necessarily the optimal vector-valued measure in the Brenier-Benamou formulation of $W_2$.  However, since an optimal measure is a potential, previous work proposed minimizing  (1) subject to the additional constraint $v_t = \nabla \phi$ for some $\phi: \mathbb R^d \rightarrow \mathbb R$.

The main contributions of this work are

1. It shows existence of a minimizer of (1) with the additional potential constraint, closing a gap in the literature.

2. It provides a counterexample that a fixed point of the iterative rectification with this constraint does not lead to an optimal coupling in the $W_2$ minimization problem, in particular this may not happen when $\mu_0$ and $\mu_1$ have disconnected support, revealing a gap in the literature.

3. It provides affine invariances and other properties of minimizers of (1) with and without the potential constraint, and compares them with the corresponding properties of the optimal vector field in the Brenier-Benamou formulation of $W_2$. The manuscript also derives explicit solutions  for the minimizers of (1) without the potential constraint in the Gaussian measure case.

Strengths and weaknesses

The classic $W_p$ metric possesses several properties that make it attractive as a loss function in machine learning and other data-driven problems, as an alternative to the standard $L^p$ norms and other mismatch functionals.  Computing $W^p_p$ is a convex optimization problem that in general does not have a known closed form solution, and this problem is challenging in high dimensional settings.  One important exception is when $\mu_0$ and $\nu_0$ are Gaussian, the $W_2$ distance has a closed form solution  (we also have a closed form expression for $W_p$ when $\mu$ and $nu$ are measures over 1D domains).

On fundamental level,  this work improves our understanding of when iterative rectification may lead to optimal $W_2$ transport couplings and optimal velocity fields, revealing a gap in the existing literature on this subject. I believe the paper provides a collection of interesting and original results at the intersection of generative modeling and optimal transport. However, if I understood the paper correctly, it does not affirmatively prove any results when  iterative rectification may lead to optimal $W_2$ transport couplings, it only provides counterexamples.

The present paper is a theoretical paper with limited numerical experiments involving toy data. From application oriented perspective, my main question is how relevant these results given that when the distributions are Gaussian, which is typical for diffusion models, the $W_2$ distance, the corresponding optimal transport map, and the optimal density and optimal vector field in the Brenier-Benamou formulation of $W_2$ have  explicit closed form solutions.  More generally are rectified flows, relevant in the context of any problems involving real data and non-Gaussian models?  Are there any references / compelling applications of rectified flows with models that are not Gaussian (and therefore optimal couplings/flows can be computed in closed form) to real world problems/ datasets?

Questions

Main questions

You seems to be suggesting that the claim of (6) has not been proven in [30] or there were gap in their proof, but if I understood the manuscript correctl you only provide a counterexample when (6) does not hold . Are you able to prove some version of (6) when $X_t$ has full support, i.e. when iterative rectification leads to optimal coupling. Do you need to lower bound $\mu_0$ or $\mu_1$ by some constant strictly above zero uniformly on the domain to achieve this?

Minor comments

* Were any new conceptual ideas/proof techniques developed in the proofs in the present manuscript?

* To contextualize this work better, why was $W_2$ used instead of $W_p$ with some other value of $p$?

* page 3 paragraph entitled "Iterative Rectification" - does the beginning of this paragraph define $\mathcal R(X_0, X_1)$? If so, it should be made clear how $\mathcal R$ is defined. Similarly, the next paragraph should be a little clearer  how $\mathcal R_p$ is defined.

Limitations

Yes, except as noted above.

Paper formatting concerns

None.